

# Biomass Burning Emissions Analysis Based on MODIS AOD and AeroCom Multi-Model Simulations

Mariya Petrenko[1,2], Ralph Kahn[3,2], Mian Chin[2],
Susanne E. Bauer[4], Tommi Bergman[5], Huisheng Bian[2], Gabriele Curci[6], Ben Johnson[7], Johannes W. Kaiser[8,9], Zak Kipling[10], Harri Kokkola[11,12], Xiaohong Liu[13], Keren Mezuman[14,4], Tero Mielonen[11], Gunnar Myhre[15], Xiaohua Pan[1,2], Anna Protonotariou[16], Samuel Remy[17], Ragnhild Bieltvedt Skeie[15], Philip Stier[10], Toshihiko Takemura[18], Kostas Tsigaridis[19,4], Hailong Wang[14], Duncan Watson-Parris[20], Kai Zhang[14]

[1]Earth System Science Interdisciplinary Center (ESSIC), University of Maryland, College Park, Maryland 20740, USA
[2]Earth Science Directorate, NASA Goddard Space Flight Center, Greenbelt, Maryland, 20771, USA
[3]Laboratory for Atmospheric & Space Physics, The University of Colorado Boulder, Boulder CO 80303, USA
[4]NASA Goddard Institute for Space Studies, New York, NY, USA
[5]Climate System Research, Finnish Meteorological Institute, Helsinki, Finland
[6]Dipartimento di Scienze Fisiche e Chimiche – CETEMPS, Universita' degli Studi dell'Aquila, Via Vetoio, 67100 Coppito - L'Aquila Italy
[7]Met Office, Exeter, UK
[8]Climate and Environmental Research Institute NILU, Norway
[9]Atmospheric Chemistry Department, Max Planck Institute for Chemistry, 52072 Mainz, Germany
[10]Department of Physics, University of Oxford, Oxford, UK
[11]Atmospheric Research Centre of Eastern Finland, Finnish Meteorological Institute, Kuopio, Finland
[12]University of Eastern Finland, Department of Technical Physics, Kuopio, Finlands
[13]Department of Atmospheric Sciences, Texas A&M University, College Station, USA
[14]Atmospheric, Climate, and Earth Sciences Division, Pacific Northwest National Laboratory, Richland, Washington, USA
[15]CICERO-Center for International Climate Research, Oslo, Norway
[16]National and Kapodistrian University of Athens, Faculty of Physics, Athens, Greece
[17]HYGEOS, Lille, France
[18]Research Institute for Applied Mechanics, Kyushu University, Fukuoka, 816-8580, Japan
[19]Center for Climate Systems Research, Columbia University, NY, USA
[20]Scripps Institution of Oceanography and Halıcıoğlu Data Science Institute, University of California San Diego, La Jolla, CA, USA

*Correspondence to*: Ralph Kahn (ralph.kahn@lasp.colorado.edu)

**Abstract.** We assessed the performance of 11 AeroCom models in simulating biomass burning (BB) smoke aerosol optical depth (AOD) in the vicinity of fires over 13 regions globally. By comparing multi-model outputs and satellite observations, we aim to: (1) assess the factors affecting model-simulated, BB AOD performance using a common emissions inventory, (2) identify regions where the emission inventory might underestimate or overestimate smoke sources, and (3) identify anomalies that might point to model-specific smoke emission, dispersion, or removal, issues. Using satellite-derived AOD snapshots to constrain source strength works best where BB smoke from active sources dominates background aerosol, such as in boreal

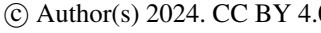



forest regions and over South America and southern-hemisphere Africa. The comparison is poor where the total AOD is low, as in many agricultural burning areas or where background, non-BB AOD is high, such as parts of India and China. Many inter-model BB AOD differences can be traced to differences in model-assumed values for the mass ratio of organic aerosol to organic carbon, the BB aerosol mass extinction efficiency, and the aerosol loss-rate. The results point to the need for increased numbers of available BB cases for study in some regions, and especially to the need for more extensive, regional-

to-global-scale measurements of aerosol loss rates and of detailed microphysical and optical properties; this would better constrain models and help distinguish BB from other aerosols in satellite retrievals. More generally, there is the need for additional efforts at constraining aerosol source strength and other model attributes with multi-platform observations.

## 1 Introduction

Aerosol particles emitted from biomass burning (BB) play a significant role in both regional climate and air quality, and in

aggregate, can contribute significantly to direct and indirect aerosol climate forcing (e.g., Andreae et al., 2004; Bowman et al., 2009; Ichoku et al., 2012; Lelieveld et al., 2015; Lu et al., 2018; Randerson et al., 2006a; Solomos et al., 2015).  One of the challenges of representing BB smoke in models that assess their environmental impacts is adequately characterizing the strength of BB sources.

Several approaches have been taken to estimate smoke source strength. A widely used set of methods involves calculating the product of burned area, available fuel load, combustion completeness and emission factors of primary aerosols and precursor gases (Seiler and Crutzen, 1980), where the latter three quantities are determined, to the extent possible, from field observations. Burned area is derived from reflectance changes in satellite imagery (e.g., Giglio et al., 2006, 2009; Roy et al., 2008; Soja et al., 2004; Wiedinmyer et al. 2011) or deduced, with some assumptions, from space-based 4-micron brightness

temperature anomaly (designated fire radiative power or FRP) measurements (van der Werf et al. 2006; Randerson et al. 2012). Other approaches exploit correlations between FRP and combustion rate (Wooster et al. 2005; Kaiser et al. 2009). The active fire (FRP)-based methods are generally more sensitive to small fires than those relying on burned area estimates; however, FRP is more affected by observational gaps due to sampling frequency limitations and cloud cover, whereas burned area can be assessed for some time after active burning has ceased (e.g., Randerson et al. 2012).


Observations of FRP combined with the aerosol optical depth (AOD) of the smoke plume itself and/or the difference between the 4 and 11-micron brightness temperatures, all obtained from the NASA Earth Observing System's MODerate resolution Imaging Spectroradiometer (MODIS) instruments, have also been used directly to estimate smoke emissions (Ichoku and Kaufman 2005; Ichoku and Ellison, 2014; Konovalov et al., 2014; Sofiev et al., 2009; Wooster et al., 2005).  One

implementation of this approach (Ichoku and Ellison, 2014) uses the plume AOD and area, divided by the advection time, estimated from the apparent length of the plume in the MODIS imagery and a wind speed obtained from a reanalysis product,





and correlates this quantity with the FRP for multiple cases to derive ecosystem-specific coefficients, which, when multiplied by the observed FRP for individual fires, yield a smoke mass emission estimate.

Inverse modeling has also been applied in efforts to characterize aerosol source strength from large-scale maps of AOD (e.g., Dubovik et al., 2008; Vermote et al., 2009; Chen et al., 2019). With this approach, a version of an aerosol transport model is effectively run in reverse, initialized with a regional or global AOD distribution, to trace back to the locations and strengths of the aerosol sources. However, this approach requires all other aspects, e.g., transport, removal, chemical transformation, source location, and non-BB aerosols, to be adequately represented in the model.


Bottom-up inventories are derived from laboriously collected information about primary and secondary aerosol sources, both anthropogenic and natural, to estimate the resulting aerosol accumulation in the atmosphere (e.g., Liousse et al., 2010; Petrenko et al., 2012; Schultz et al., 2008; Seiler & Crutzen, 1980; Van der Werf et al., 2010; Wiedinmyer et al., 2011). This approach has been an essential tool for approximating aerosol loading for times prior to global satellite observations and continues to be

a key resource for estimating regional aerosol amounts and types, but it suffers from limited knowledge about source properties, as well as unknown sources that would be missing altogether.

Not surprisingly, there are significant discrepancies among the different estimates of BB aerosol source strength (e.g., Petrenko et al. 2012, henceforth P2012; Pan et al., 2020). In an effort to bring additional satellite-based constraints to bear on smoke

source-strength estimates globally, P2012 adopted a forward-modeling approach that made explicit use of known smoke source locations, and compared model-derived estimates of aerosol loading for varying aerosol source strength with satellite-derived AOD rather than using top-of-atmosphere brightness temperature itself to characterize smoke source strength. Region-specific summaries of the relationships between smoke emission rates used in the model and MODIS-retrieved snapshots of AOD for individual plumes were provided. In particular, in the P2012 study, the GOCART model (Chin et al., 2002) was initialized

with varying BB sources as specified by a number of widely used smoke source emission inventories including the Global Fire Emission Database version 3 (GFED3) (Van der Werf et al., 2010; Randerson et al., 2012). The model was sampled at the time closest to that of satellite overpass, and the near-source AOD of the model was compared with that derived from coincident MODIS observations.  One key observation from this study is that the model simulated AOD bias within a given geographic region is systematic, such that the model overestimated, underestimated, or approximately agreed with the observed AOD

snapshots for nearly all plume cases within that region. This indicated that it might be possible to apply region- and/or biome-specific adjustment factors to the emission inventories to bring the model into agreement with the observations.

Petrenko et al. (2017; henceforth P2017) greatly expanded the database of smoke cases in P2012, and refined the model-observation comparisons (1) by using scaled AOD reanalysis values from the Modern-Era Reanalysis for Research and

Applications (MERRAero) to fill AOD in those parts of plumes too optically thick to derive AOD from MODIS observations



and in areas obscured by clouds, (2) by distinguishing to the extent possible the emitted BB aerosol from background aerosol generated by other sources, and (3) by assessing qualitatively the effect of small-fires based on emissions from the GFED4.1s database (Randerson et al., 2012; 2017; Van der Werf et al., 2017) to account for fires too small to be detected by the standard, satellite-based methods used for GFED3. This analysis showed that the overall approach works best when both the total AOD

and the BB fraction of total AOD are high, which occurs primarily for evergreen or deciduous forest fires.  Ambiguities arise when either the background AOD is comparable to or larger than the BB contribution, generally in heavily polluted regions such as northern India and eastern China, or when the total AOD is low, which can occur in regions of sparse vegetation or agricultural burning.

The P2012 and P2017 studies looked only at results from the GOCART model, which provided a consistent set of results that were relatively straightforward to interpret in terms of emission source strength. However, those studies did not address the uncertainties associated with a range of underlying model assumptions that are not constrained by the choice of BB emission source strength alone. The current study expands upon this earlier work, by examining the behavior of 11 models that are part of the AeroCom community.  The results highlight some of the leading model assumptions, not well-constrained by

measurements, that affect model-simulated AOD even when the emission strength is specified.

The AeroCom community has a long history of performing multi-model experiments in which certain factors are controlled among the model runs, and comparative analysis yields insights into the impact of different model assumptions and parameterizations (e.g., Kinne et al., 2006; Textor et al., 2006; Huneeus et al., 2011; Tsigaridis et al., 2014; Bian et al., 2017;

Kim et al., 2019; Gliss et al., 2021, Zhong et al., 2022).  These efforts have produced a great many insights into the factors affecting model performance and have made it possible to isolate model-specific factors from issues associated with the external constraints.  Following this tradition, and as part of the larger AeroCom Phase III multi-model experiment effort, the Biomass Burning experiment presented here is designed to compare coincident samples of AeroCom model AOD simulations with satellite-derived plume AOD, primarily from MODIS, to constrain smoke source strength.  In an associated effort to

improve BB source modeling, the satellite-derived smoke plume injection heights from the NASA Earth Observing System's Multi-angle Imaging Spectroradiometer (MISR) (Val Martin et al., 2018) are being used to initialize the models; the resulting smoke dispersions are being compared with those obtained using the nominal model injection heights, as has been done for a few individual models previously (e.g., Vernon et al., 2018; Zhu et al., 2018) and is currently being evaluated with AeroCom models as well (Pan et al., 2022; manuscript in preparation). The overall aim of these AeroCom BB experiments is to assess

the smoke source strength inventories and injection heights widely assumed in models, in the context of global satellite-derived constraints, and in addition, to identify any model-specific issues that arise from the comparisons.

The current paper reports the results of the AeroCom Biomass Burning Source-Strength Experiment, for which the same BB emissions inventory from the Global Fire Emission Dataset version 3.1 (GFED3.1) is used in all participating models. The



model-simulated results are evaluated region-by-region with the MODIS smoke plume reference database developed in P2012 and P2017. In the process, we also refined the set of geographic regions to better match areas showing distinct smoke behavior as well as to correspond to the extent possible with the biomass burning regions defined by the GFED (Giglio et al., 2006). The objectives of this study are: (1) to assess, and as much as possible quantify, the accuracy and diversity of the AeroCom-model-simulated BB AOD using a common emissions inventory, (2) to identify regions where the emission inventory might
underestimate or overestimate smoke sources, based on the comparison between multi-model outputs and the satellite observations, and (3) to identify any anomalies that might point to individual, model-specific issues regarding smoke emission, dispersion, or removal, based on the model-measurement analysis.

Section 2 describes the model experiment, reviews the model characteristics, and summarizes the techniques used to analyze
the results.  Section 3 presents the key results globally and by region and biome. However, the model simulations yield different results even when initialized with the same emissions, so this section also identifies cases where individual models appear as outliers and identifies the range of model assumptions for which better observational constraints are needed.  The paper concludes with a summary of results and provides a review of the strengths and limitations of the approach.

## 2 Experiment Overview and Analysis Approach

**2.1. AeroCom model experiment**

For the AeroCom-III BB Source-Strength experiment, eleven models submitted sufficient diagnostics to perform the analysis presented here. Information about model structure, dynamical processes, and BB emission injection height for this experiment is listed in Table 1. Additional information on sources of aerosols other than BB smoke, and assumed particle microphysical properties for the 11 models, is listed in supplemental tables S1 and S2. The models represent a diversity of spatial resolutions,
parameterizations, and assumed particle sizes and properties. For example, horizontal resolution ranges from about 0.5×0.625˚ (GEOS) to 4˚×5˚ (GEOS-CHEM), and vertical layers from 30 (CAM5) to 85 (HadGEM). Meteorological fields were obtained from different reanalysis products. Although the modelers were asked to distribute BB emission within the model boundary layer, some models chose to prescribe other BB emission injection altitudes. For example, CAM5 injected smoke evenly within the lowest 1 km, ECMWF-IFS-CY45R1 distributed the amount within the lowest 2 km, OsloCTM2 incorporated a
geographically varying injection height with maximum height of 5 km, and ECHAM6-SALSA injected the smoke between 0 and 5 km, depending on the ecosystem.





**Table 1: Characteristics of participating models**

| Model name (version) | Kind of model, resolution (lat × lon × lev) | Met fields used | Convection (dry and moist) | Boundary layer definition | BB EIH (for this project) | References |
|---|---|---|---|---|---|---|
| CAM5 (v5.3) | GCM nudged by reanalysis (1.9°×2.5°×30) | ECMWF ERA-Interim reanalysis | Park and Bretherton (2009); Zhang and McFarlane (1995) | Diagnostic TKE-based 1st-order K diffusion scheme with entrainment parameterization (Park & Bretherton, 2009). | Evenly distributed within 0-1 km | (X. Liu et al., 2012; Ma et al., 2013; Neale et al., 2012; Wang et al., 2013; Wiedinmyer et al., 2006; K. Zhang et al., 2014) |
| ECHAM6-SALSA (6.1) | GCM nudged by reanalysis (1.9°×1.9°×31) | ECMWF ERA-Interim reanalysis | Nordeng (1994) | Equation 3, Stevens et al. (2013) | Ecosystem-specific emission profiles from 0 to 6 km | Stevens et al. (2013); Bergman et al. (2012); Kokkola et al. (2018) |
| ECMWF-IFS (CY45R1) | GCM, nudged by reanalysis (T255 x 60, i.e. ~80 km horizontal resolution) | MACC reanalysis (Inness et al., 2013) | Bulk mass scheme (Bechtold et al., 2014; Tiedtke, 1989) | Diagnostic following Troen and Mahrt (1986) | 2 km | Flemming et al., 2015; "IFS Documentation," n.d.; Morcrette et al., 2009; Remy et al, 2019 |
| GEOS5 | GCM, replay with MERRA2 (0.5°×0.625°×72) | MERRA2 (Gelaro et al., 2017) | Dry: Resistent-in-series (Wesely, 1989); Moist: Relaxed Arakawa-Schubert convection scheme (Moorthi & Suarez, 1992) | Shear-based component of the turbulent kinetic energy (TKE) (McGrath-Spangler & Molod, 2014) | Evenly distributed within the BL | Bian et al. (2009) Chin et al. (2002) Colarco et al. (2010) |
| GEOS-CHEM (v9-02) | Off-line CTM (4°×5°×72) | GEOS5-DAS (Rienecker et al., 2008) | non-local PBL, (Holtslag & Boville, 1993). Moist: Relaxed Arakawa-Schubert convection scheme (Moorthi & Suarez, 1992) | Recalculated internally as a function of atmospheric stability (Lin et al., 2008; Lin & McElroy, 2010) | Evenly distributed within the BL | Liu et al. (2001) |



| GISS ModelE MATRIX | GCM nudged by reanalysis (2°×2.5°×40) | NCEP reanalysis horizontal winds 6 hourly | Schmidt et al., 2014 (and references therein) | Dynamic PBL | Evenly distributed within the BL. | Schmidt et al., 2014; Bauer et al. 2008 |
|---|---|---|---|---|---|---|
| GISS ModelE OMA | GCM nudged by reanalysis (2°×2.5°×40) | NCEP reanalysis horizontal winds 6 hourly | Schmidt et al., 2014 (and references therein) | Dynamic PBL | Evenly distributed within the BL. | Schmidt et al., 2014; Koch et al., 2006;2007; Bauer et al., 2007a, 2007b; Miller et al., 2006a, 2006b; Tsigaridis et al., 2013 |
| GOCART (5 rev. 32) | Off-line CTM (1°×1.25°×72) | MERRA (Rienecker et al., 2011) produced with GEOS5-DAS | | provided by GEOS5 | Evenly distributed within the BL. | Chin et al. (2000, 2002, 2007, 2009, 2014) |
| HadGEM (3) | Atmosphere-only GCM nudged by reanalysis (1.25°×1.875° ×85) | ECMWF ERA-Interim reanalysis | BL mixing scheme based on Lock et al. (2000). Mass flux based on Gregory and Rowntree (1990), Derbyshire et al. (2011) | Diagnosed from stability profile (non-local scheme accounting for moist parcel ascent) | Evenly distributed from surface to 3 km. | |
| OsloCTM2 | Off-line CTM (2.8°×2.8°×60) | ECMWF IFS forecasts for year 2008 | Wet removal in grid boxes with convective precipitation from IFS (Berglen et al., 2004) | PBL hight given in the IFS data. | EIH from project RETRO, between 0 and 5 km (Schultz et al., 2007) | Myhre et al.(2009); Skeie et al. (2011) |
| SPRINTARS (5.5) | GCM nudged by reanalysis (1.125°×1.125° ×56) | ECMWF ERA-Interim reanalysis | Takemura et al. (2009; 2000, 2002, 2005) | Takemura et al. (2009; 2000, 2002, 2005) | Evenly distributed within sigma level larger than 0.74. | Takemura et al. (2009; 2000, 2002, 2005) |



The year 2008 was selected as the "benchmark year," with prescribed daily biomass burning emission from GFED3.1 for this study. Among the reasons for selecting this emission dataset and simulation time period were to examine the robustness of the
analysis done for the single-model simulation presented in P2012 and P2017 and to evaluate the multi-model results with hundreds of satellite-observed cases compiled in these previous studies (summarized in section 2.3). Other aerosol emissions, including emissions from desert dust, fossil fuel combustion, and other anthropogenic and natural sources, were determined by the individual models.

In this study we are using model output from two simulations: a control run (BB1) with all sources including prescribed daily BB emissions from GFED3.1, anthropogenic emissions from a number of external emission inventories (Table S1) chosen by the modeling groups, and natural sources such as dust and sea salt calculated by models, and a run with the same sources but with no BB emissions (BB0). The difference between BB1 and BB0 allows the BB contribution to be isolated from other contributions to aerosol load. In addition to these baseline simulations, the models performed three perturbation runs with the
GFED3.1 daily emissions multiplied by factors of 0.5 (BB0p5), 2 (BB2), and 5 (BB5), respectively, to create an ensemble of four runs where multiples of GFED3.1 represent a range of possible emission estimates for the same fires. The models were run for the full year, preceded by a three-month "spin-up."

## 2.2 The GFED BB Emissions

GFED is one of the most widely used BB emission inventories in the global modeling community. It is also continuously
updated to include the latest findings in BB emission development studies (Giglio et al., 2013; Randerson et al., 2012, 2017). At the time when the AeroCom BB experiment was proposed, GFED3.1 (Mu et al., 2011; Randerson et al., 2012; Randerson et al, 2013; Van der Werf et al., 2010) was the latest GFED version available. It was, therefore, used for the model runs performed for the current study, although the later version, GFED4, was available after the model runs were performed (the emissions were compared with those from GFED3 by e.g., Giglio et al., 2013, and referred to by Petrenko et al., 2017).
GFED3.1 provides daily biomass burning emissions of CO, $SO_2$, $NO_x$, $NH_3$, VOCs (volatile organic carbon), BC (black carbon), and OC (organic carbon). The map of 2008 annual GFED3.1 emission of OC, the most abundant primary aerosol species emitted from fire, is shown in Figure 1a.

The OC emissions provided by GFED3.1 had to be converted to organic aerosol mass (OA, a.k.a. organic matter or OM) by multiplying OC by an OA/OC ratio that is based on information from various observations. However, this ratio depends on
the chemical age of OA, the particular OA species, and environmental conditions; it therefore can have a wide range of values, typically from a little over 1 to well above 2 (e.g., Aiken et al., 2008). As a result, although the same OC emissions are used, the primary OA from BB emissions varies among the models by nearly a factor of 2. Specifically, the OA value reflects the choice of OA/OC ratio incorporated in individual models, with OsloCTM2 and SPRINTARS having the highest values (2.6) and GISS, GEOS, HadGEM3 and ECHAM6-SALSA having the lowest (1.4), as illustrated in Figure 1b. In contrast, the BC
emissions from all models are nearly identical (Figure 1c). Figure 1c also displays that a primary peak occurs in July-August,



when burning tends to favor northern mid-to-high latitudes and the southern subtropics; secondary peaks occur in December-January and in April, when burning occurs preferentially in the northern hemispheric tropics (see Figure 2).

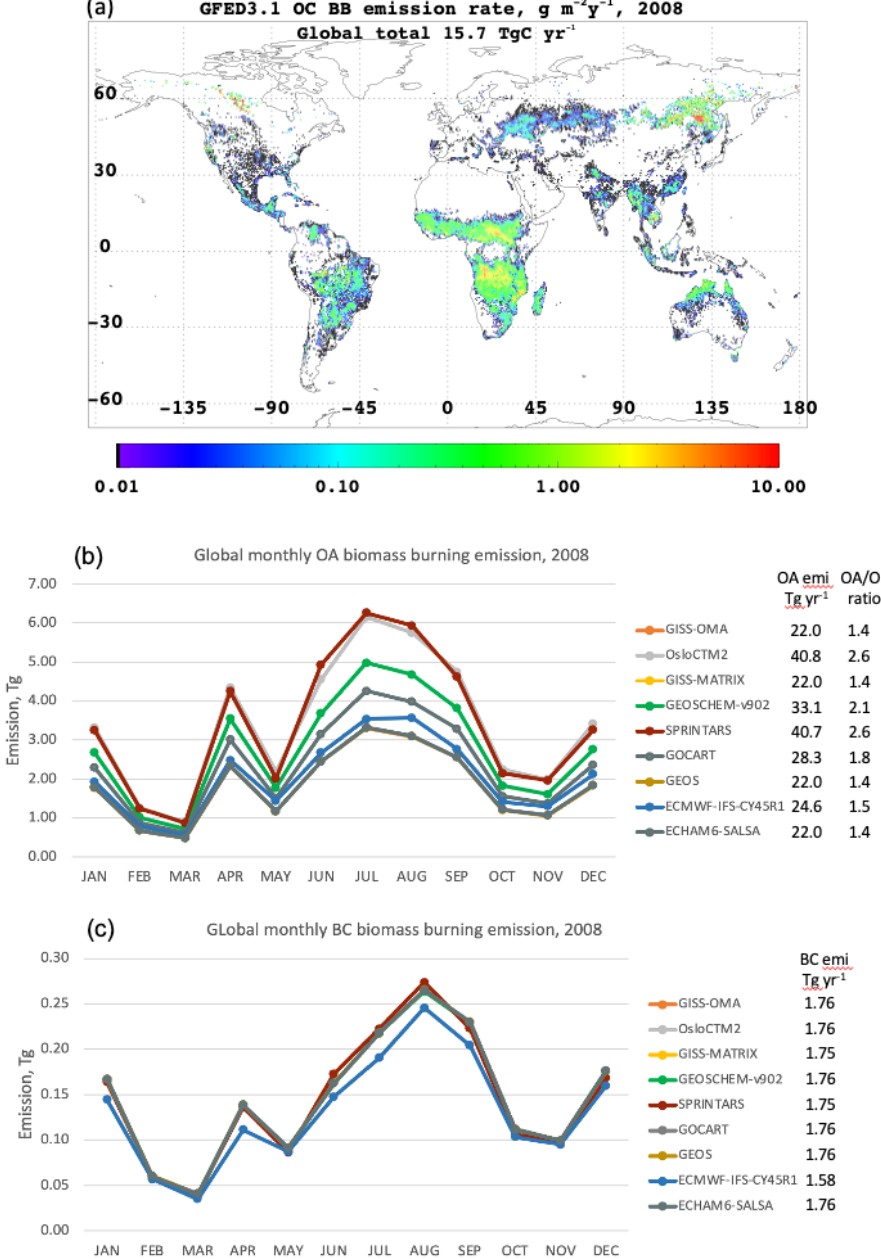

**Figure 1: Global biomass burning emissions of carbonaceous aerosols. (a) Annual emission of OC in 2008 from GFED3.1, (b)**
**monthly OA emissions implemented in 9 participating AeroCom models converted from OC with the OA/OC ratio of model's choice (listed in the legend to the figure), (c) same as (b) but for BC. (Note: colored lines in 1b and 1c can overlap for models with identical emissions).**





## 2.3 The MODIS BB plume AOD Dataset

We use the MODIS Collection 6 Level 2 AOD data from the Terra and Aqua satellites as the key observational dataset to evaluate and constrain the models. The MODIS BB plume AOD dataset was introduced and refined in P2012 and P2017, respectively. Here, 447 fire/smoke cases in different biomass burning regions that fall within the benchmark year of 2008 are selected as the reference observational dataset, from about 900 identified in P2017. The main criteria for selecting BB cases are detailed in P2012 and P2017; briefly, these include: (1) plumes with at least one linear dimension of 100 km, to be useful for global modeling studies with fairly coarse resolution of 1° or larger (Table 1), (2) a coordinated pattern of elevated AOD, (3) a visible smoke plume in the satellite imagery, and (4) a fire signal in the MODIS thermal anomalies product (MOD14). The locations and seasonality of the cases in the database are shown in Fig. 2.

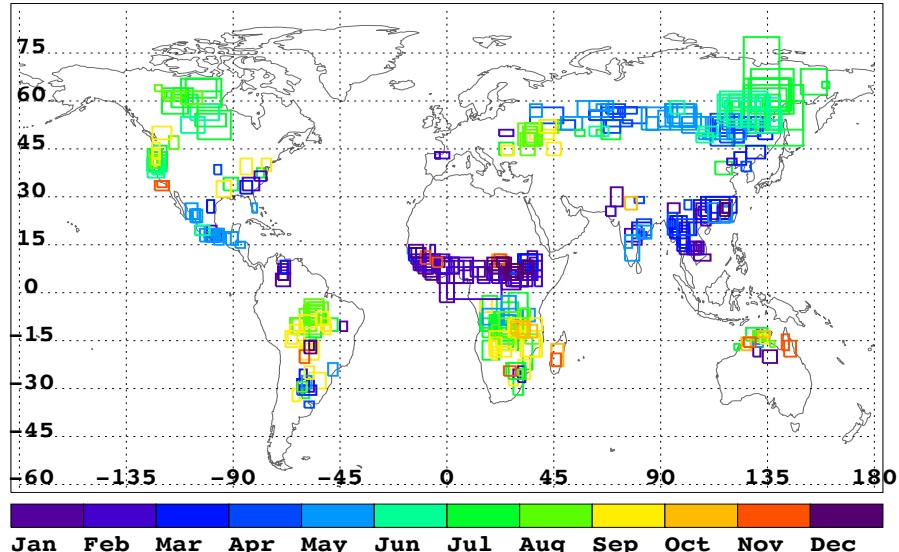

**Figure 2: Locations and months of the fire case boxes in this study.**

To estimate BB AOD from MODIS, we first estimate the background AOD value, i.e., AOD from non-biomass burning sources, for each MODIS pixel from a statistical analysis of 16 years of available MODIS Terra data during pre-burning-season months, and then subtract these values from the total observed AOD, as in P2017. Missing MODIS AOD retrievals within the plumes are filled with MERRAero reanalysis values (Buchard et al., 2015), scaled to retrieved MODIS AOD values in immediately surrounding locations where both MODIS and model values are available. (Details are presented in P2017.) Note that fire activity in Alaska, Indonesia, and South Australia was rather weak in 2008, so no cases were specified in these regions.



## 2.4 Biomass burning regions

Based on the analysis in P2017 and the regional characteristics of fires, our analysis in the current study focuses on the same geographical regions. To better associate our analysis with other biomass-burning-focused studies (e.g., Giglio et al., 2006; Rabin et al., 2015; Pan et al., 2020; Mezuman et al., 2020), we adopt the region names used by GFED (Giglio et al., 2006), and assign our cases with these regions (Figure 3). In addition, we further divide the BOAS region into eastern (BOAS_E) and western (BOAS_W) subregions, and CEAS into eastern (CEAS_E) and western (CEAS_W) parts, mainly to account for

observed differences in burning patterns within the broader GFED regions. In total, 13 regions/subregions are included in the current study. The regions are shown in Figure 3, and the BB cases within each region are displayed as symbols, with different symbol styles assigned to distinctive groups based on the degree of concurrence between the satellite and model BB estimates, as discussed in the next section.

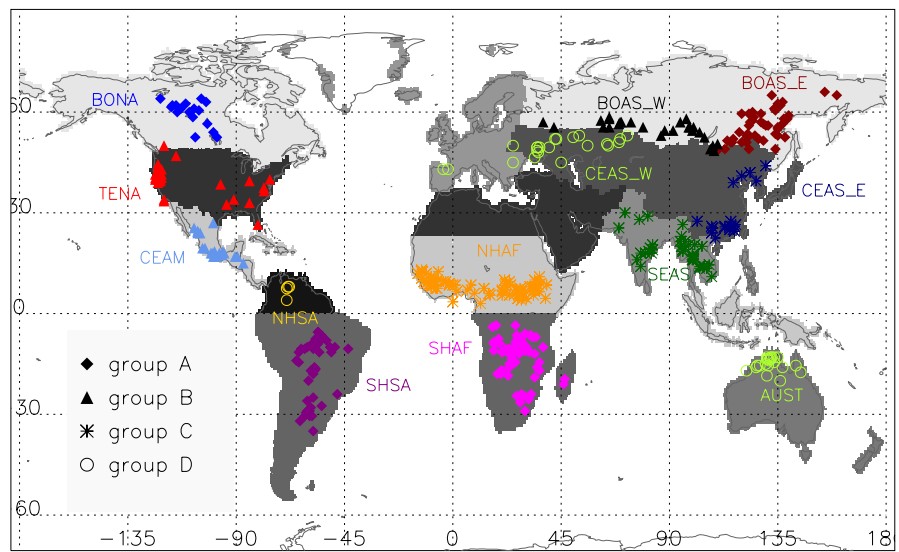

**Figure 3: The 13 regions with the BB cases in each region. BONA = Boreal North America, TENA = Temperate North America, CEAM = Central America, NHSA = Northern Hemisphere South America, SHSA = Southern Hemisphere South America, NHAF = Northern Hemisphere Africa, SHAF = Southern Hemisphere Africa, BOAS_W = Boreal Asia West, BOAS_E = Boreal Asia East, CEAS_W = Central Asia West, CEAS_E = Central Asia East, SEAS = Southeast Asia, AUST = Australia. Symbols for BB cases mark the group (A, B, C or D) that the BB region belongs to. The groups of BB regions are explained in section 3.2.**

## 2.5 Comparing average values

In order to compare BB emissions and BB AOD between models, we obtain model BB AOD by subtracting results of the no-BB-aerosol simulation (BB0) from the control run with all emissions (BB1). The method for obtaining MODIS BB AOD is detailed in P2017 and is briefly summarized in section 2.3 above. We then use the instantaneous model output closest in time to the satellite observation to calculate case-average values. As each rectangular case box is defined by a set of latitude-

longitude coordinates, the model output was sampled to include all the grid boxes with the centers of which falling within the case box. Average values from MODIS and the models were then compared over the area of the box.





When comparing values in further analysis, we calculate average values in the following ways:

- **Case box average AOD** (also for BB AOD, load, loss, and extinction efficiency) is the arithmetic mean of all AOD values within a case box. For BB AOD, we first subtracted the background AOD (a fixed, pre-determined, case-specific value for MODIS, and the no-BB run for models) from all AOD pixels in the case box to obtain BB AOD, then set any negative BB AOD values to 0, and then averaged BB AOD over the case box.
- **Regional average** is the simple arithmetic mean of all average case values for cases assigned to the region.
- **All case average AOD** (or BB AOD) is the simple arithmetic mean of all average case AOD (or BB AOD) for all 447 cases in the study.
- **Global monthly values** include all grid boxes weighed by area, averaged over a month (used for model-to-model comparisons only)
-

When working with variables that represent ratios of values (such as model-to-satellite AOD ratios, loss frequency, or mass extinction efficiency), the robust mean is often used to exclude any values falling beyond 4 standard deviations of the mean to discard outliers. This approach ensures that, in regions with very low AOD values, the ratios of a few very small numbers do not skew the regional averages unreasonably. This treatment rejects 0-10% of the case values from making the regional averages.

## 3 Results

### 3.1 Comparisons between MODIS and model BB AOD cases over biomass burning regions

Figure 4 shows the spatial distribution of simulated BB AOD relative to the estimated MODIS BB AOD described in section 2, covering all the individual cases for each model. The models are ordered from the highest to lowest overall BB AOD (when all cases are averaged). Many common features among the models relative to MODIS appear in Figure 4. For example, the models report generally lower BB AOD than the MODIS estimates, except in some cases in central and southern Africa. However, most do fall within 50% (ratio between 0.67-1.5) of the MODIS-derived values over the boreal region of North America (BONA), southern and parts of central Africa, northern Venezuela/Columbia, and northern Australia. The model BB AOD simulations tend to be much lower over the U.S., Mexico, western boreal region of Asia, central and southeast Asia, China, and India, generally by factors of 5 to >10.

These model-to-MODIS BB AOD ratios are quantified in Table 2 for all models and all regions. To make discerning regional patterns easier, table cells were colored according to the color scheme in Fig. 4. These color clusters in Table 2 emphasize the spatial patterns described above. The regions are further collected into groups A, B, C, and D as discussed in the next section. Deeper dive into absolute values of BB variables for each model in each region is available in the supplemental figure S3.



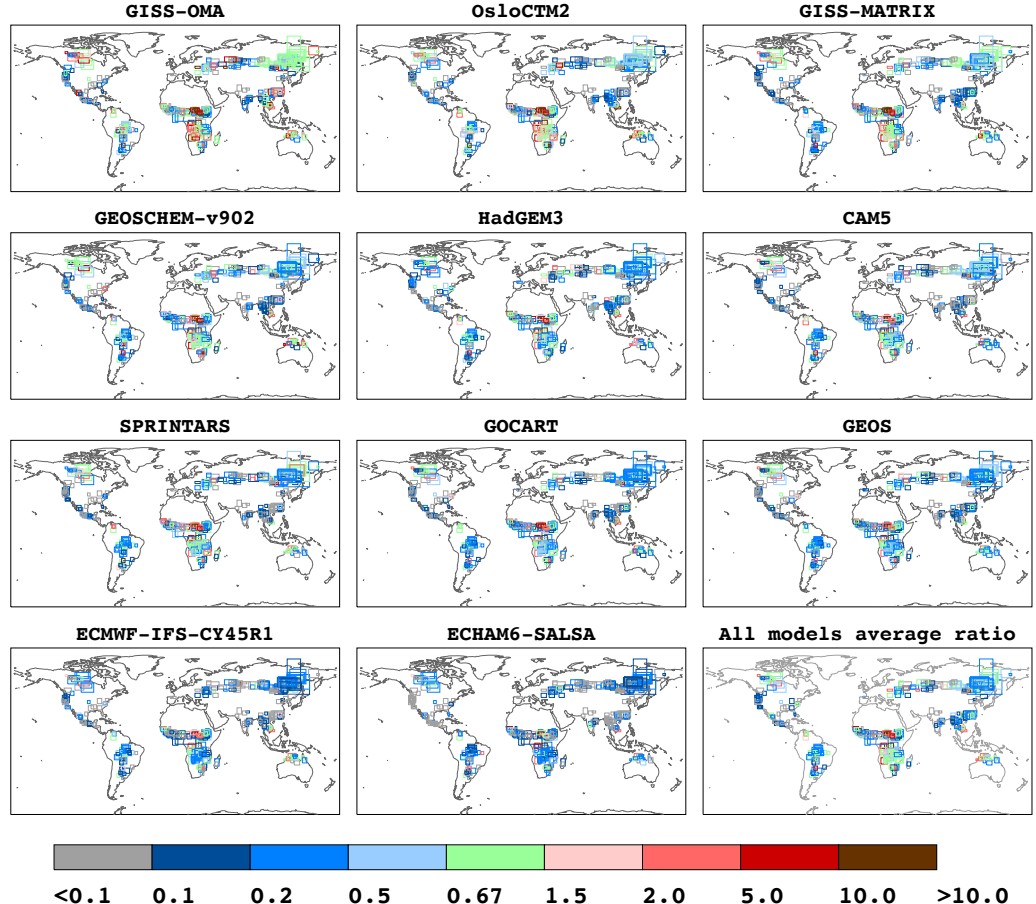

**Figure 4a: Ratio of model-simulated BB AOD (from model experiment BB1) to the BB AOD derived from MODIS for all individual fire cases for each individual model, and (last panel) the multi-model average of there ratios for all study cases.**

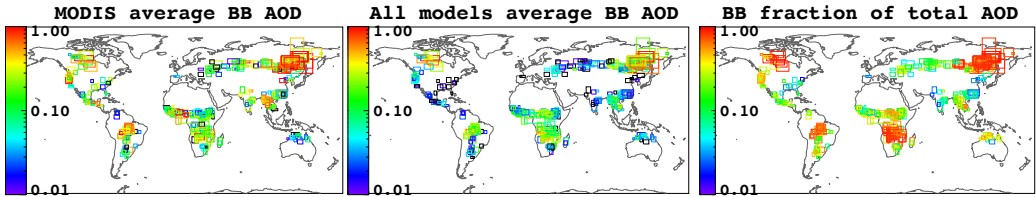

**Figure 4b: BB AOD derived from MODIS for reference (left) and averaged across all models (middle), and BB AOD fraction of total AOD averaged across all models (right) for all study cases.**



**Table 2: Ratios (r) of model calculated BB AOD to MODIS-derived BB AOD for cases within each of the 13 regions. Colors illustrate the bias of individual model relative to MODIS. The means also tabulated. Regions are further grouped into A, B, C, and D based on the degree of the agreement between multiple models and MODIS to help discussion.**

| GROUP | REGIONS | GISS-OMA | OsloCTM2 | GISS-MATRIX | GEOSCHEM-v902 | HadGEM3 | CAM5 | SPRINTARS | GOCART | GEOS | ECMWF-IFS-CY45R1 | ECHAM6-SALSA | Region Mean |
|---|---|---|---|---|---|---|---|---|---|---|---|---|---|
| A | BONA | 2.57 | 1.33 | 1.67 | 0.84 | 0.93 | 1.45 | 0.76 | 1.21 | 1.49 | 0.53 | 0.62 | 1.22 |
| A | SHAF | 2.15 | 1.89 | 1.25 | 0.99 | 0.86 | 0.81 | 0.78 | 0.69 | 0.62 | 0.50 | 0.41 | 1.00 |
| A | SHSA | 0.67 | 0.76 | 0.53 | 0.54 | 0.49 | 0.44 | 0.53 | 0.41 | 0.39 | 0.34 | 0.23 | 0.48 |
| A | BOAS_E | 0.99 | 0.75 | 0.74 | 0.69 | 0.61 | 0.56 | 0.73 | 0.51 | 0.56 | 0.30 | 0.21 | 0.61 |
| B | BOAS_W | 0.46 | 0.37 | 0.48 | 0.48 | 0.21 | 0.16 | 0.24 | 0.24 | 0.29 | 0.12 | 0.13 | 0.29 |
| B | CEAM | 0.12 | 0.21 | 0.13 | 0.12 | 0.11 | 0.10 | 0.08 | 0.11 | 0.11 | 0.11 | 0.04 | 0.11 |
| B | TENA | 0.16 | 0.17 | 0.15 | 0.11 | 0.21 | 0.13 | 0.10 | 0.12 | 0.09 | 0.12 | 0.04 | 0.13 |
| C | NHAF | 1.02 | 1.92 | 1.35 | 1.02 | 0.85 | 1.17 | 1.16 | 1.05 | 1.09 | 0.56 | 0.44 | 1.06 |
| C | SEAS | 0.28 | 0.24 | 0.18 | 0.18 | 0.16 | 0.14 | 0.10 | 0.12 | 0.12 | 0.10 | 0.10 | 0.16 |
| C | CEAS_E | 0.65 | 0.13 | 0.17 | 0.14 | 0.20 | 0.07 | 0.07 | 0.07 | 0.11 | 0.06 | 0.16 | 0.16 |
| D | CEAS_W | 0.63 | 0.56 | 0.46 | 0.43 | 0.18 | 0.30 | 0.32 | 0.31 | 0.35 | 0.13 | 0.12 | 0.34 |
| D | NHSA | 1.44 | 2.20 | 0.84 | 1.08 | 1.32 | 1.85 | 2.02 | 1.02 | 0.80 | 0.63 | 0.52 | 1.25 |
| D | AUST | 1.65 | 2.07 | 0.91 | 1.54 | 1.36 | 1.23 | 1.04 | 1.15 | 0.80 | 1.09 | 0.61 | 1.22 |
| Model mean | | 0.98 | 0.97 | 0.68 | 0.63 | 0.58 | 0.65 | 0.61 | 0.54 | 0.52 | 0.35 | 0.28 | 0.62 |

| r < 0.1 | 0.1 ≤ r < 0.2 | 0.2 ≤ r < 0.5 | 0.5 ≤ r < 0.67 | 0.67 ≤ r ≤ 1.5 | 1.5 < r ≤ 2 | 2 < r ≤ 5 | 5 < r ≤ 10 | r > 10 |
|---|---|---|---|---|---|---|---|---|

## 3.2 Separating BB regions into different groups

To comprehensively compare multiple variables for 11 models over 13 regions, we developed a multi-factor region-comparison approach. For example, in P2017 we considered the magnitudes of total MODIS and model AOD, biomass burning fraction of total AOD, and model/satellite BB AOD ratio to assess how effectively our method of estimating source-strength by comparing modeled and measured AOD can be used in different BB regions.

We begin here by stratifying the regions into groups, according to several observation-based criteria that reflect the level of confidence in our ability to identify the MODIS and the model BB AOD components. The criteria for grouping regions are:

1. ***Total AOD from MODIS***. MODIS AOD retrieval uncertainties are much lower when the AOD is above about 0.1 (Levy et al., 2013). So, under the conditions when MODIS AOD is sufficiently high, if the total AOD discrepancy



between the models and MODIS is large, it is likely an issue with the models, such as emission strength or model processes. This provides important regional information in the context of the current study. If the total AOD from MODIS is low, then the relative uncertainty in the estimated MODIS BB AOD is expected to be high.

2. ***Biomass burning AOD fraction from MODIS when total AOD is high***. If the BB AOD fraction (fBB) is also high (i.e., the estimated "background," non-BB AOD fraction is low), we have greater confidence in the MODIS BB AOD
obtained by subtracting the estimated background AOD from total AOD. Otherwise, the estimated MODIS BB AOD is more uncertain.

3. ***Total AOD and BB AOD from models.*** If both total AOD and BB AOD fraction from models are relatively high, we are more certain that our constraints can be applied to assess the biomass burning emission source strength, as intended. Otherwise, more issues related to the model simulation of BB and other (background) aerosol types (e.g.,
pollution, dust, etc.) complicate interpretation of the results.

4. ***Low small-fires correction***. Small fires can be missed by satellite observations and as a result, can be lacking in emission inventories that use primarily satellite methods to detect fires. Small fires would therefore not be included in models that use such inventories. Based on our assessment in P2017, we found that applying a "small fire correction" to the original GFED3 emissions affects the resultant BB AOD in some regions. Those regions where the
impact of small fires is minimal are better suited for our approach for constraining BB emissions.

Figure 5 is a flowchart showing the process we used to assign regions to particular groups, using the first three criteria listed above. Criterion #4 is used as an additional uncertainty consideration. Overall, the 13 biomass burning regions in Figure 3 are associated with Group A, B, C, or D based upon the process described in Figure 5.




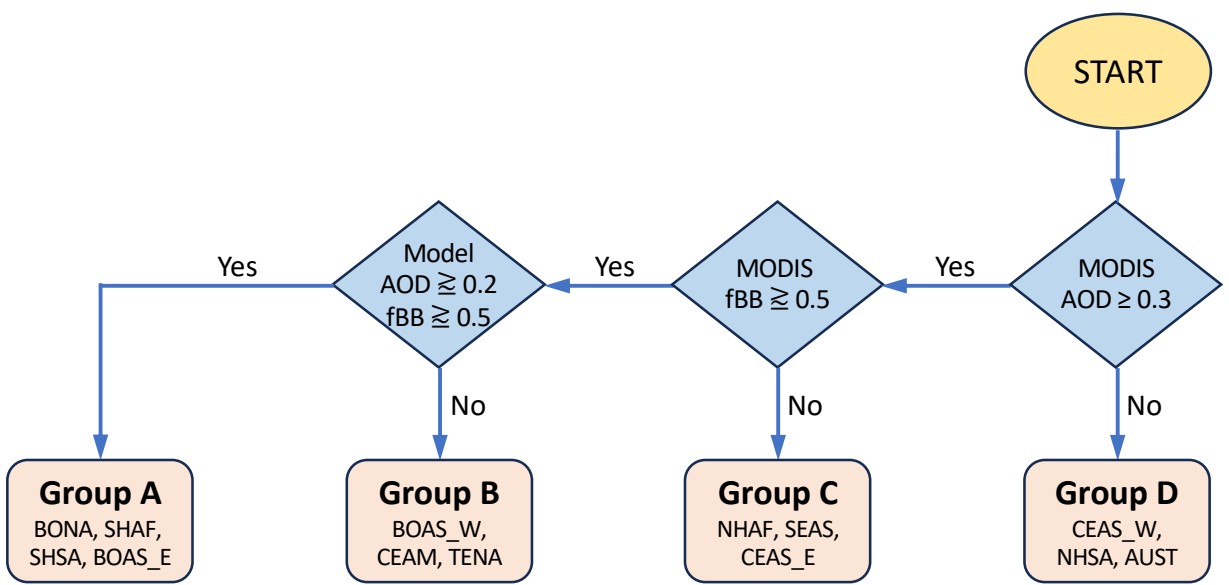

**Figure 5: Flow chart of the procedure separating the 13 biomass burning regions into four groups with distinctive characteristics in biomass burning intensity, fraction of smoke AOD w.r.t. total AOD (fBB), and differences between the quantities from MODIS and the multi-model mean. Regions in each group and the characteristics are shown in Fig. 6.**


Quantitative representation of regional all-model means for these criteria is provided in Table 3. To make discerning regional patterns of factor magnitudes in Table 3 easier, we again color-coded each entry for the parameters given in each column, based empirically on parameter values that correspond to varying degrees of agreement between measured and modeled BB AOD. For example, MODIS total AOD greater than or equal than 0.3 is colored green, AOD lower than 0.3 is yellow. As such,

the color codes represent qualitative criteria favorable for applying the satellite AOD to constrain emission source strength in the models, green being more favorable, and yellow – less. Mean values of all available model averages for the region also report the diversity, calculated as the ratio of standard deviation to the mean of all model means for this region, expressed as percentages. The last two columns show the importance of small fire correction in each region based on the approach described in P2017. The closer the number to unity, the less significant this correction is, meaning the emissions in the regions are fairly

accurately estimated already and not many small fires were missed in creating the emission inventory.





**Table 3: Multi-factor comparison of BB regions**

| Group | GFED name | P2017 name | # cases | MODIS total AOD | MODIS fBB | total AOD all-models mean (diversity) | models fBB[a] | small fires correction important (p2017, T1) | model/MODIS BB AOD ratio all-models mean (diversity) |
|---|---|---|---|---|---|---|---|---|---|
| A | BONA | Canada | 17 | 0.34 | 0.57 | 0.31 (40%) | 0.86 ( 8%) | 1.00 | 1.25 (49%) |
| | SHAF | SAfrica | 66 | 0.31 | 0.57 | 0.23 (41%) | 0.72 (14%) | 0.80 | 1.01 (58%) |
| | SHSA[b] | SAmerica | 45 | 0.33 | 0.68 | 0.18 (26%) | 0.49 (18%) | 0.86 | 0.48 (32%) |
| | BOAS_E | Russia (E) | 47 | 0.65 | 0.73 | 0.38 (40%) | 0.75 (11%) | 0.67 | 0.61 (37%) |
| B | BOAS_W | Russia (W), Europe | 27 | 0.37 | 0.57 | 0.16 (41%) | 0.23 (44%) | 0.55 | 0.30 (47%) |
| | CEAM | LAmerica | 23 | 0.35 | 0.56 | 0.10 (35%) | 0.20 (25%) | 0.67 | 0.11 (37%) |
| | TENA | WUSA + EUSA | 37 | 0.40 | 0.65 | 0.09 (21%) | 0.37 (21%) | 0.92 | 0.12 (30%) |
| C | NHAF | NCAfrica | 79 | 0.30 | 0.36 | 0.43 (37%) | 0.31 (42%) | 0.63 | 1.08 (37%) |
| | SEAS[c] | SEAsia + India | 37 | 0.45 | 0.52 | 0.25 (19%) | 0.21 (32%) | 0.74 | 0.16 (40%) |
| | CEAS_E | China | 20 | 0.58 | 0.24 | 0.27 (32%) | 0.06 (56%) | 0.63 | 0.16(109%) |
| D | CEAS_W | Europe | 22 | 0.19 | 0.31 | 0.14 (31%) | 0.18 (30%) | 0.82 | 0.36 (45%) |
| | NHSA | N of SAmerica | 4 | 0.06 | 0.14 | 0.08 (45%) | 0.26 (49%) | region undefined before | 1.24 (48%) |
| | AUST | NAustralia | 22 | 0.06 | 0.58 | 0.10 (35%) | 0.36 (31%) | 1.00 | 1.21 (36%) |
| | | | | < 0.3 | <0.5 | < 0.2 | <0.5 | | <0.2 or >5 |
| | | | | | | | | | |
| | | | | >= 0.3 | >=0.5 | >= 0.2 | >=0.5 | 0.67 - 1.5 | 0.67-1.5 |

[a]fBB is fraction of total AOD attributed to biomass burning aerosol.

[b] Total model AOD and models fBB are rounded up to 0.2 and 0.5 respectively, putting SHSA in group A.

[c] Even though MODIS fBB in SEAS is higher than the cutoff threshold for group C, the complex aerosol mixture in this region makes our confidence in MODIS background AOD values (and thus in MODIS fBB of 0.52) rather low, and the combination of fairly high model AOD and low BB AOD fraction in the models puts this region in group C.





**3.3 Broad view of MODIS and model comparisons in biomass burning regions and groups**

We present a broad view of MODIS and model comparisons by region in Figure 6. The general model behavior is represented by the multi-model mean values of AOD and BB AOD. We show in Figure 6 top panel the total AOD (stacked blue-shaded bars) as well as background and BB AOD from MODIS (light and dark blue bars, respectively) and the corresponding multi-model mean values (light and dark red bars) averaged for cases that fall within each region. The 13 regions and divided into

the four regional groups, designated earlier as A, B, C, and D based on physical criteria. Also shown are the BB AOD fractions for MODIS and for the model means in blue and red lines, respectively. In addition, ratios of model mean total AOD, background AOD, and BB AOD to the corresponding MODIS quantities are shown in the bottom panel of Figure 6 with solid, dashed, and dot-dashed lines, respectively.

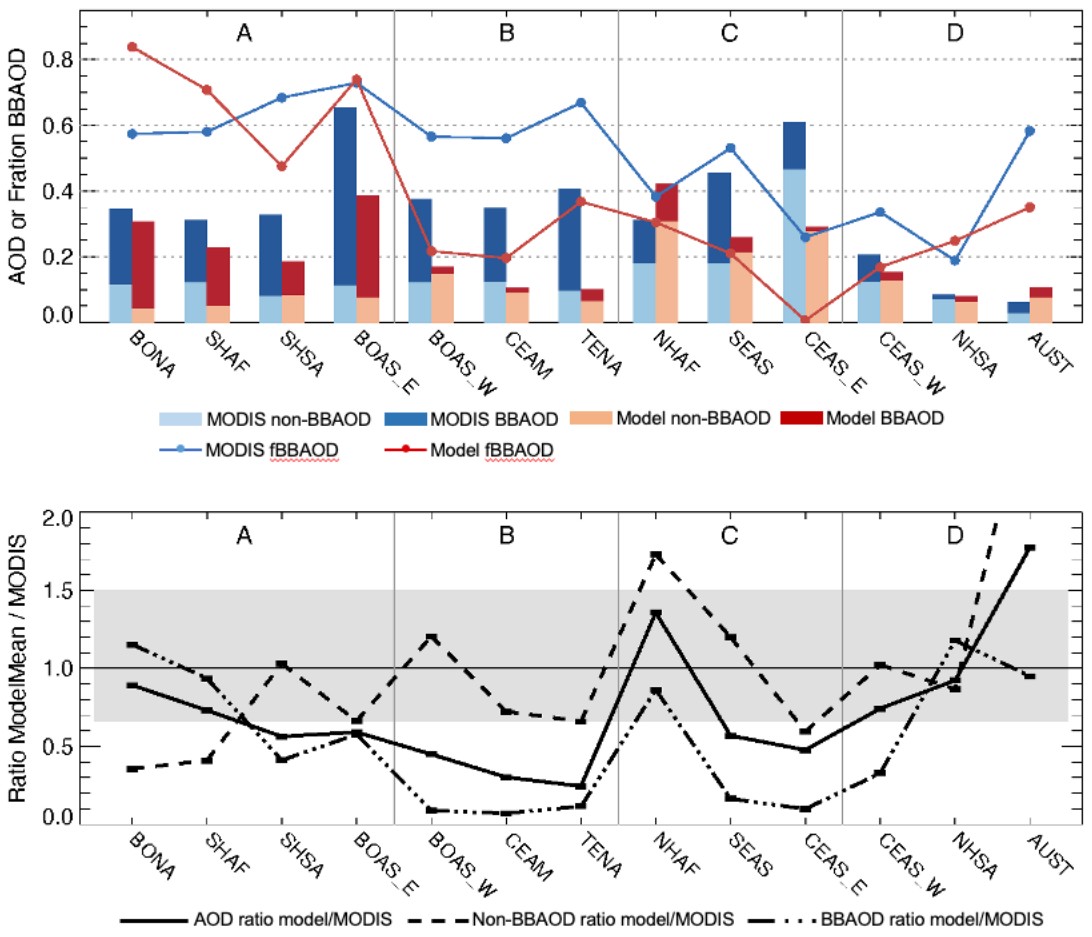

**Figure 6: Top: Total AOD from MODIS (stacked blue-shaded bars) and from multi-model mean (stacked red-shaded bars), their corresponding non-BB background AOD (light colors) and BB AOD (dark colors), and their BB AOD fractions (lines), averaged for cases in each of the 13 regions grouped by A, B, C, and D (see Fig. 5 and text). Bottom: Ratios of model mean to MODIS for total AOD (solid line), non-BB background AOD (dashed line), and BB AOD (dot-dashed line). The light gray shade indicating the range of model to MODIS ratio (R) within 50% (0.67 ≤ R ≤ 1.5).**




Four regions (BONA, SHAF, SHSA, and BOAS_E) fall into Group A, where AOD and BB AOD fractions from both MODIS and model means are generally high (AOD ≥ 0.3 for MODIS and ≧ 0.2 for the model mean, BB AOD fraction ≧ 0.5 for both MODIS and the model mean). Tree cover dominates in these regions, with few other aerosol sources and typically well-defined

fire plumes or major burning events (see also P2012 Table 4, and P2017 Fig 1a).

Unlike Group A, model mean AOD and BB AOD are both dramatically lower than MODIS in Group B by factors of 5-10 for AOD (solid line, bottom panel in Fig. 6) and around 20 for BB AOD (dot-dashed line, bottom panel in Fig. 6). However, for the Group B regions, the non-BB background AOD between MODIS and the model mean agrees, to within 50%, with the ratio

of model/MODIS for non-BB AOD = 0.67-1.2 (dashed line, bottom panel in Fig. 6). Given the high AOD and > 0.5 BB AOD fractions based on MODIS, and agreement between MODIS and model on background AOD, we are more confident to suggest that the GFED3.1 BB emission is systematically low or has missed sources in the group B regions; a high bias in MODIS total AOD and low bias in our MODIS background subtraction are also possible but is less likely. Fires in TENA (USA) are mostly in needle-leaf evergreen tree-covered areas, with many fire cases in complex mountainous terrain where both satellite AOD

retrieval and model transport simulations are difficult. BOAS_W is mostly covered by sparce herbaceous and shrub vegetation, plus cultivated and managed areas; CEAM has a combination of fire cases in broadleaved evergreen tree cover and cultivated and managed lands. Small fires that are likely to be missed in the GFED inventory are common in sparsely vegetated, cultivated, and managed lands (Randerson et al., 2012).

Although total AOD from MODIS in Group C is of similar magnitude as that in Groups A and B, the fraction of BB AOD is much lower. Regions in Group C contain BB cases with a variety of tree and shrub/grass/cropland vegetation types but are heavily influenced by either dust in NHAF or high pollution in SEAS and CEAS_E, making the MODIS background subtraction as well as the model-simulated BB contribution to the total AOD highly uncertain for this group. Meanwhile, the non-BB background AOD is higher for both MODIS (0.18-0.46) and model mean (0.21-0.31) than for any other group. Such

high non-BB AOD fractions reduce the confidence in our BB source-strength estimates in these regions.

In Group D, MODIS total AOD is the lowest among all groups, at 0.06-0.20, and the BB signal is very weak, resulting in estimated BB AOD at 0.015-0.08. As such, small errors in any aspect of the MODIS retrievals can produce large relative uncertainties. Among the regions in Group D, AUST fire cases are mostly in areas with deciduous shrub-cover, and CEAS_W

is dominated by cultivated and managed lands. There are only four cases for NHSA, so statistics for this region are not robust. Although the model mean AOD and BB AOD generally agree with the corresponding MODIS values within a factor of 2, the confidence in our source-strength estimations in the Group D regions is limited because of the low signal in the observations.





From the above analysis, we can reach a few conclusions about biomass burning emissions of GFED3.1 used by the models. The biomass burning emissions are most likely to be realistic in Group A regions, but they should be increased by a factor of 2-10 in Group B regions for the models to be in line with the satellite BB AOD based on the agreement between model and satellite data for the background non-BB AOD. Model results from BB5 run yield a model-to-MODIS BB AOD ratio of around 0.7 for TENA, 0.6 for CEAM and 2.5 for BOAS_W, suggesting that multiplying aerosol emissions by 2 in BOAW_W and almost 10 for TENA and CEAM would make model and MODIS BB AOD comparable. Because of the high non-BB (background) AOD fractions in Group C and the low total AOD and BB AOD in Group D, we do not have confidence to draw any conclusion about biomass burning emission strength over regions within these groups.

## 4 Model diversity

### 4.1 Multi-model means of BB OA quantities in each region

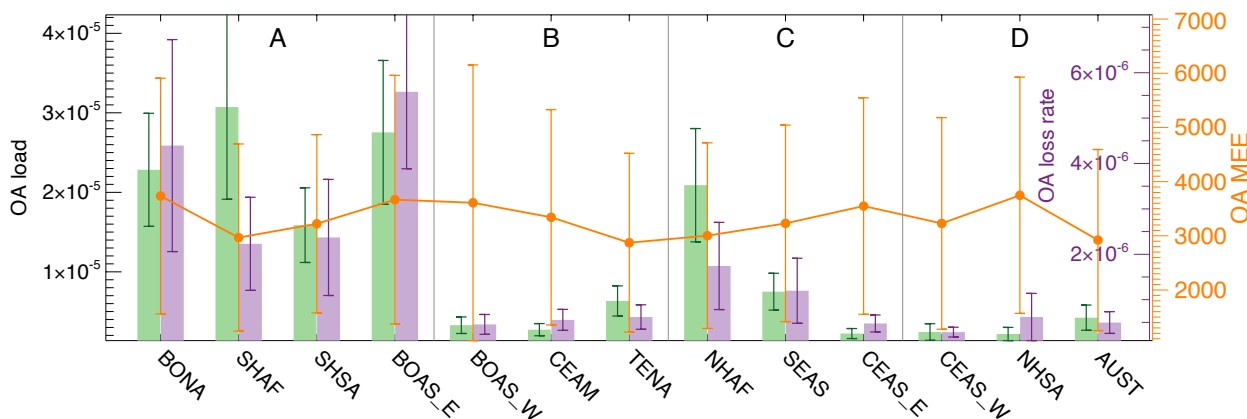

**Figure 7: Multi-model mean OA load (green bars, left Y axis), multi-model mean OA loss rate (purple bars, right purple Y axis), and multi-model mean mass extinction efficiency of OA (orange points and whiskers, right orange Y axis) averaged for cases in each of the 13 regions grouped by A, B, C, and D (see Fig. 6 and text). Error bars show standard deviations of the multi-model means, respectively.**

We show in Figure 7 the multi-model mean quantities of BB OA mass load (green bars), loss rate (purple bars), and the mass extinction efficiency (MEE, orange line) that converts model BB OA mass to BB OA AOD, in each region. As expected, more and/or larger fires in the regions of group A correlate with higher BB aerosol loads. The residence time, related to OA removal from the atmosphere in each region, can be estimated from dividing the load by the loss rate; from the relative heights of the green and purple bars for each region in Figure 7, we can estimate the different residence times of OA among regions. For example, in group A, OA residence time in boreal regions BONA and BOAS_E (higher purple bars than green) is shorter than that in SHAF and SHSA (higher green bars than purple), reflecting the differences in mass balance of smoke aerosol emission,





deposition, and transport fluxes in each region. On the other hand, the multi-model mean OA MEEs, calculated as the ratio of BB AOD to BB load here, are similar across the regions in all groups (3000-4000 $m^2$/kg) despite large differences of BB OA mass or load in these regions. MEE is an intensive aerosol property that depends on particle refractive indices, size

distributions, particle density, and hygroscopic growth under ambient conditions. However, despite the region-by-region similarity in the mean values, MEE diversity among individual models is remarkable, as seen from the large MEE standard deviations in each region, resulting from the combination of differences in the optical (refractive indices) and microphysical (particle size, hygroscopicity) properties used in each model. Details of individuals model values by region are provided in the supplemental Fig. S4.

**4.2 Diversity in atmospheric processes and BB optical properties among models**

Fundamentally, the column AOD reported by the models is derived from the aerosol mass loading in the atmosphere and the efficiency with which radiation is scattered and absorbed by the mass of a given aerosol species present, i.e., the MEE, under ambient atmospheric conditions. Globally, total aerosol mass load within the models is the result of total source (including primary aerosol emission and secondary aerosol production), and removal processes (including dry and wet deposition and

chemical loss). These factors control aerosol amount and lifetime in the atmosphere. On the other hand, the MEE depends upon aerosol composition, size distribution, particle density, refractive indices, and hygroscopicity that usually depend upon the ambient relative humidity.

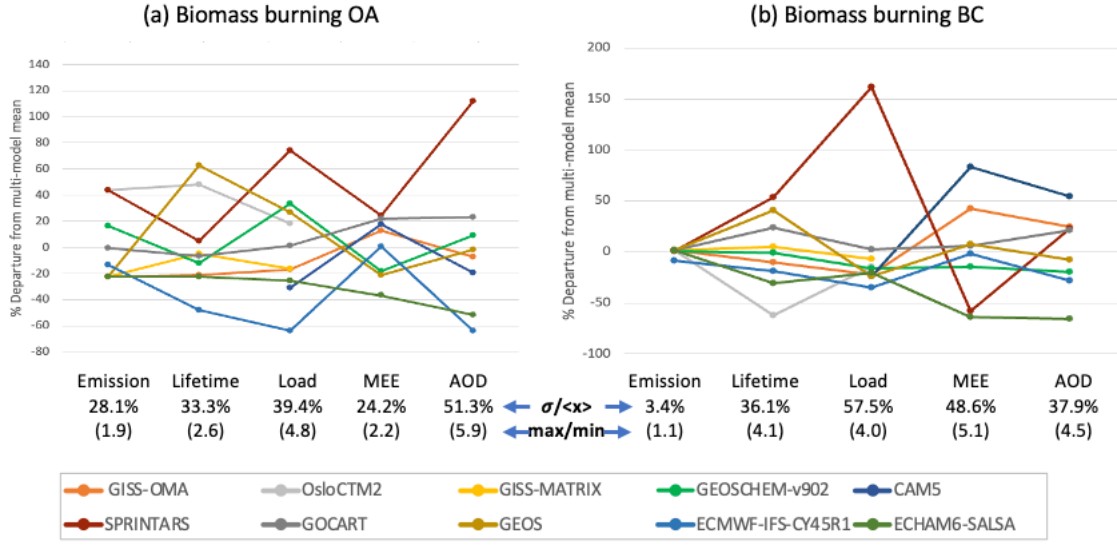

**Figure 8: Differences among model-simulated key parameters determining the (a) OA AOD and (b) BC AOD from biomass burning**
**sources expressed as the percentage departure of each model from the multi-model mean values. The quantities are derived from global mean values for 2008. Some parameters are missing from a few models. The model diversity of each parameter, defined as % of standard deviation/multi-model mean, for each parameter is listed under the corresponding parameter. The spread of values, represented by the ratio of the largest of the model values for the corresponding parameter to the smallest is given in parentheses under corresponding diversity.**




Although the transport processes that affect aerosol spatial distribution might explain some of the model differences regionally, comparisons among the global values of the key quantities determining the BB AOD can shed light on the model diversity that underlies regional differences relatively independent of the transport. Here, we compare the individual-model global values of five key biomass burning OA and BC aerosol quantities for 2008 in Figure 8: emission, lifetime, atmospheric mass loading,

MEE, and BB AOD, expressed as the percentage departure of individual models from the multi-model mean, with the numerical value of the overall spread given below each parameter label. (Note that in taking the global mean of all the BB variables, we subtracted the BB0 from the BB1 model runs, which effectively compares model characteristics in general, not just those assessed for the specific regions and cases considered elsewhere in this study.)

As discussed with Fig. 1b and 1c, although all models use the same BB emissions for BC and OC from GFED3.1, the different OA/OC ratios chosen by individual models result in nearly a factor of two difference in OA emissions, producing an inter-model diversity of 28% essentially at emission (Figure 8a), in contrast with the nearly identical emissions of BC (diversity of 3.4%, Figure 8b). Higher BB emissions generally lead to higher BB AOD globally, but this is only part of the story, as the diversity of speciated BB AOD among the models (51% for OA, 38% for BC) is much greater that of their corresponding BB

emissions. Among all models, SPRINTARS tends to produce higher BB AOD than other models in the study set, whereas ECMWF-IFS-CY45R1 and ECHAM6-SALSA are consistently at the low end. Although OsloCTM2 and GISS-OMA BB AOD are also among the highest of the models overall (see Fig. 4 and Table 2), their BB aerosol composition is internally mixed, making estimating OA AOD fraction of BB AOD not possible. Thus, BB OA AOD and consequently MEE for these two models are not included in Fig.8. The differences can be traced back to the disparity of the OA BB emission rates to begin

with: OA emission from SPRINTARS and OsloCTM2 is 80% higher than that from ECMWF-IFS-CY45R1 and ECHAM6-SALSA because of the different OA/OC ratios assumed (see Figure 1b). On the other hand, some model behaviors are difficult to explain. For example, globally, SPRINTARS and OsloCTM2 have the same OA emission rates, but the OA load from SPRINTARS is 60% higher despite having a 50% shorter lifetime than that from OsloCTM2. One way this could occur is if the SPRINTARS model has more efficient OA removal.


Another factor that adds diversity to models' treatment of OA is the simulation of secondary organic aerosol. Among the models in this study, all emissions shown in Fig.8 are for primary OA (POA), but BB OA AOD includes both primary and secondary organic aerosol (SOA) in the CAM5, OsloCTM2, and GEOSChem models. In the attempt to work with total OA output provided by the models, whether the model includes SOA simulation or not, these three models also include SOA in

their load and loss estimates. (Note that some models such as GOCART and GEOS have SOA produced from non-biomass-burning sources that are included in the total OA but not in BB OA.)



The diversity of BC BB AOD is less than that of BC load and MEE, partly due to some compensating factors that can be seen in Figure 8b. For example, SPRINTARS and GOCART have the same BC BB emission and the same BC BB AOD, but the

BC load from SPRINTARS is 2.5 times larger and the BC MEE is 2.5 times smaller than the corresponding values for GOCART. Also notably, CAM5 has the highest MEE, making its simulated BC BB AOD the highest among the models, despite moderately low BC BB mass loading. In the model results, AOD = load (kg m$^{-2}$) x MEE (m$^2$ kg$^{-1}$), load (kg m$^{-2}$) = lifetime (days) x loss rate (kg m$^{-2}$ day$^{-1}$). The emission rate was prescribed to be the same for all models, and by conservation, the loss rate should roughly equal the emission rate on a global annual basis. So, the diversity in BC BB AOD is driven by the

differences between a) BC lifetime, governed by the removal processes, and b) MEE, determined by the particle size distributions and hygroscopic growth with ambient relative humidity, neither of which is well constrained, due primarily to a lack of adequate observations. Moreover, another AeroCom study by Brown et al. (2021) involving many of the same models, suggest that BB aerosols in the global aerosol models are too absorbing. In the case of OA (Figure 8a), the diversity of emission rates contributes to the diversity of model-simulated OA BB AOD.


In summary, although consistency among the models does not necessarily indicate accurate representation of smoke plume properties and behavior, model diversity does provide at least a lower bound on uncertainty. Individual, significant outliers point to areas where specific questions about model assumptions might be asked, and more generally, observations are clearly needed to better constrain loss mechanisms and MEE.

**5 Discussion**

The multi-model diversity illustrated in Section 4 above highlights uncertainties in the models, particularly in the MEE, loss frequency, and OA/OC mass ratio, as well as uncertainty in the MODIS background subtraction, that limit the confidence with which any method combining satellite-retrieved AOD with model simulations can constrain source strength, or other model attributes. However, having identified these limitations, we can at least apply the method in places offering the best conditions

for assessing smoke source strength with this approach, i.e., the Group A and possibly Group B regions, with appropriate consideration of the uncertainties involved in these areas.

In our earlier study (P2017), we demonstrated that working with BB AOD rather than total AOD is essential to evaluate BB source strength. This requires MODIS total AOD to be separated into AOD from BB and non-BB aerosol. Until now, we are

not aware of a reliable method for isolating the BB component in MODIS AOD. The approach we developed and described in P2017, and also applied in the present study, is rather crude. The multi-model analysis presented here underlines the limitations of this method, especially in regions with high non-BB aerosol fractions, such as several regions in groups C and D. This calls for the application of satellite measurements with more reliable BB AOD separation methods, such as having multi-angle (e.g.,



Kahn & Gaitley, 2015; Junghenn Noyes et al., 2022) and possibly polarization as well as multi-spectral sensitivity (e.g.,

Dubovik et al., 2019) in global remote-sensing measurements.

Background AOD subtraction for BB AOD measurement estimation is likely to improve once tighter constraints on satellite-retrieved particle properties (e.g., Junghenn Noyes et al., 2020) become more widely available. Also, current global models may best be used to compare coarser-resolution variables, e.g., averaged over larger areas and over weeks or months, rather

than comparing individual events. Our study indicates that focusing on snapshots of single events might require obtaining a larger sampling of cases in some regions and/or having models offering finer spatial resolution. Also, there might be other, novel ways to run models that would better isolate specific sources, and thus improve inter-model and model-measurement comparisons. As applied here, the approach works best for large, well-defined smoke plumes in low-background environments, primarily deciduous and evergreen forest, especially in boreal regions. We assigned these regions to Groups A and B in the

current study.

With all models significantly underestimating both total and BB AOD, but matching the MODIS background AOD values within 50%, in regions of group B (TENA, CEAM, and BOAS_W), we infer that the aerosol source-strengths input to the models from the aerosol emission inventory are most likely too low in these regions. Regions of group B contain predominantly

cultivated lands and mixed vegetation types. Both small fires and other factors likely contribute to the emissions deficit in these regions. The GFED has evolved since the model runs were performed for the current study, which used GFED3.1. Although the newer version, GFED4s includes more small fires (van der Werf et al., 2017) that should address some of the uncertainty in the emission amounts, the BB emissions from GFED4s only showed modest increases (10-40%) in the group B regions (Pan et al., 2020), certainly far from a factor of > 10 increase needed for models to match the MODIS BB AOD. In

that regard, some more aggressive BB emission estimates, such as the Quick Fire Emission Dataset (QFED), which is based on the MODIS fire radiative power (FRP). QFED is also to some extent adjusted empirically based on the MODIS AOD. As such, QFED might help to improve the model simulations of BB AOD in group B regions, since the emissions are 4-16 times higher than GFED3.1 in these regions (Pan et al., 2020). However, quantitative aspects of model performance need in addition to be assessed with different emission datasets.


The current study demonstrates that even with the same BB emissions going into the model, the resultant BB AOD varies significantly in all regions. Given the diversity in the results and the high dimensionality of the data, we could not identify any BB region or model that could be used as a benchmark for further comparison (or calibration) with confidence. In the absence of adequate observational constraints on both the particle properties and the processes involved, differences in processes and

assumptions make it possible for models with very different aerosol loads and optical properties to arrive at the similar AOD values, and conversely. For future multi-model studies, we recommend implementing common tracers into all participating models, such as a transport tracer and a removal tracer, to help isolate the causes of model diversity in these key processes.





## 6 Conclusions

We have explored in some detail the strengths and limitations of an approach to constraining wildfire smoke source strength
by comparing simulated AOD samples obtained from 11 AeroCom global models with AOD derived from space-based remote
sensing. The most meaningful results with this method are obtained for regions where individual, optically thick smoke plumes
occur and background AOD levels are low, which favors primarily boreal forests. Results are much more uncertain where total
AOD or smoke AOD is low and/or background AOD is high, associated with the Group C and D regions in the current study,
respectively. The primary factors limiting source-strength-estimation results in regions more favorable to the method include
uncertain MEE, aerosol loss frequency, and OA/OC mass ratio assumed in the models, and background AOD subtraction for
the satellite AOD values. Results in these regions will improve greatly once the requisite measurements are acquired and are
applied to constraining the models, processes not adequately addressed by current research efforts, but essential for refining
the source-strength estimation approach applied here, and far more generally, for reducing the uncertainty in modeling aerosol
effects on climate (e.g., Kahn et al., 2023).


As has also been shown in previous studies, the AeroCom consortium of modelers, especially in collaboration with the AeroSat
community that contributes measurement expertise to such investigations, together offer a broad-based, effective, and collegial
environment for pursuing advanced studies of aerosols and their impacts on climate. The great variety of assumptions,
approaches, and characteristics represented by the models participating in the current study allows us to assess the efficacy of
some key model choices.

We observe a range of biomass-burning-related results, including significant differences in atmospheric load, lifetime, assumed
particle properties, and the resulting BB AOD, among the 11 participating models, even when all models are initialized with
the same BB emissions. This often points to differences in model treatment of physical and chemical processes such as plume
injection height, aging time, removal mechanisms, and secondary aerosol formation, as well as aerosol microphysical and
optical properties such as particle size distributions, hygroscopic growth rates, and mass extinction efficiencies. At present,
these processes and properties and their associated assumptions or parameterizations are not adequately constrained by
observations in most cases. For example, higher assumed ratios of BB OA/OC (Figure 1b) are reflected in higher BB AOD for
many models (Figure 8a). More generally, some models generate lower BB AOD estimates consistently across biomes,
compared to others.

Differences also appear between model BB AOD and that estimated from MODIS AOD measurements. Some of these
differences are likely due to difficulty in distinguishing background aerosol vs. BB from specific sources in the interpretation
of MODIS data. We estimate background AOD from MODIS statistically, based on retrieved AOD for months just prior to
regional burning seasons, assessed over multiple years. Such estimates are quite uncertain, which matters primarily in regions
where other aerosol sources or aged smoke dominate, or where the total AOD is low. We associate such regions with Groups



C and D in the current study; both model and measurement estimates of BB AOD are more uncertain in these regions, resulting in poor BB source strength constraints using our method.

In summary, the observed, systematic patterns among models, and between models and estimated BB AOD from measurements, show that our approach of comparing a model AOD simulation, run in the forward direction, with satellite-retrieved BB AOD, is useful for constraining the strength of natural BB aerosol sources in some regions, a quantity for which there are few other ways to estimate empirically. It also offers an example of how satellite measurements can help place aerosol-related climate modeling on more solid ground, one major reason for acquiring such data.

**Code and data availability**

The data sets used in this work are publicly accessible and referenced in the text.

The GFED3.1 emission data set can be obtained from https://daac.ornl.gov/cgi-bin/dsviewer.pl?ds_id=1191.

Output from individual models for Phase III BB experiment are stored in the AEROCOM repository, which can be accessed by request, as described at https://aerocom.met.no/data .

MODIS data sets can be obtained from Level-1 and Atmosphere Archive & Distribution System https://ladsweb.modaps.eosdis.nasa.gov

Coordinates and descriptions of all observational study cases for 2008 are included in the supplemental table S2 in Petrenko et al, 2017, doi/10.1002/2017JD026693.

**Author contribution**

Mariya Petrenko in close collaboration with Mian Chin and Ralph Kahn designed the experiment, ran the GOCART model, coordinated data collection, analyzed data, and prepared the manuscript. The rest of the co-authors ran the models, formatted and uploaded model output, participated in data analysis discussions, and provided constructive comments on the manuscript.

**Competing interests**

The authors have the following competing interests:

Xiaohong Liu is a Senior Editor (Subject: Aerosols),

Gunnar Myhre is an Editor (Subject Areas: Aerosols, Climate and Earth System, Clouds and Precipitation, Gases, Radiation),

Philip Stier is an Editor (Subject areas: Aerosols, Climate and Earth System, Clouds and Precipitation, Radiation),

Toshihiko Takemura is an Editor (Subject Areas: Aerosols, Climate and Earth System, Clouds and Precipitation, Radiation),

Kostas Tsigaridis is an Editor (Subject Areas: Aerosols, Climate and Earth System, Gases),

Hailong Wang is an Editor (Subject Areas: Aerosols, Climate and Earth System, Clouds and Precipitation, Dynamics),



Duncan Watson-Parris is an Editor (Subject Areas: Aerosols, Climate and Earth System, Clouds and Precipitation, Radiation).

**Acknowledgements**

The work of M. Petrenko and R. Kahn on this project is supported in part by the NASA Atmospheric Chemistry Modeling and Analysis Program (ACMAP, under R. Eckman), the NASA Earth Observing System (EOS) Terra project (under K. Thome
and H. Maring), and the EOS MISR project (under D. Diner). M. Chin acknowledges NASA ISFM (under R. Eckman) and MAP (under D. Considine) programs for their support.

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
