# Peer review of "Biomass Burning Emissions Analysis Based on MODIS AOD and AeroCom Multi-Model Simulations"

_EGUsphere, 2024_

## Author Comment (AC1)

Reviewer 1
**Major:**

**GFED 3.1 is very outdated and has been shown to underestimate fires, particularly small ones. I understand that that's what was used in the multimodel study, but, at the minimum, an uncertainty analysis showing the difference between that and updated fire inventories should be included and a discussion of the limitations of GFED3.1.**

Thank you for this comment. We do realize that GFED3 has been superseded by the next version (GFED4s), which includes emissions from small fires.

We agree that GFED3.1 has been known for underestimating BB emissions in several BB regions based on many modeling studies (e.g., Petrenko et al., 2012, Pan, 2020, Carter et al., 2020); however, using GFED4s emission will not improve the agreement between model simulated and MODIS-based BB AOD and total AOD in essentially all regions, and thus will not affect the conclusions in this paper, for the following reasons:

- Comparisons between GFED3 and GFED4s products made previously by Giglio et al (2013) and van der Werf et al. (2017) show that a globally, GFED4s BB carbon emissions may be even lower that GFED3.1. As shown clearly in Pan et al., 2020 (see figure below), globally the BB emission from GFED4s is 12% lower than GFED3.1, especially in the regions where BB usually dominates (e.g. Group A regions in our study).
- For regions where GFED4s emission is higher than GFED3.1 (TENA, CEAM, CEAS, NHSA, and part of BOAS), the total emission and the increased amount are still very small (see figure below), far from sufficient to account for the significant underestimation of AOD and BB AOD from models.

[Figure]

Fig. R1. Comparisons of OC biomass burning emissions from 6 emission datasets for 14 BB regions in Pan et al., 2020.

We have added the discussion of GFED3.1 vs GFED4s BB in the Discussion section (line 550).

And even the outdated dataset, but used consistently to compare other settings, proves useful – for example, for assessing the spread of other simulated parameters, shown in Fig. 8, which is the main conclusion of this paper.

- **The models compared are also quite old. Specifically, GEOS-Chem v9-02 is super outdated. I understand that that was what was used in the intercomparison, but the science is so outdated as to raise the question – is this comparison useful now? What is the added value of a paper like this when a lot of these fire comparisons have already been done and this study is using outdated models and fire inventories? Most, if not all, of these takeaways have been shown in other work (papers by Tina Liu, Carter et al. (2020), Pan et al. like the other cite, etc.)**

We respectfully disagree on this: the main takeaways have not been highlighted previously. Specifically, we conclude that the model treatments of particle properties, and in particular, OA/OC ratio, MEE, and aerosol lifetime, are so diverse among the models that it is not possible to apply the source-strength-constraint approach consistently for even this collection of models. The particle properties in the models, even later versions, have not changed significantly since the earlier versions used in this work, as the systematic measurements required to serve as meaningful constraints have not been made. The modeler coauthors participating in the study agree with these conclusions, which call for the observations that are lacking. There are several aspects in the present study that have not been done in previous work, as stated below.

- Works by Carter et al. (2020) (thanks for pointing out this reference) and Pan et al. (2020) focus on comparing several emission inventories when used as an input to one model, whereas this study highlights the discrepancies between outputs of multiple models that use the same inventory, which shifts the focus from the inventories to the model configurations/processes and the analysis approach used for comparison.
- Specifically, we find it valuable to separate the biomass burning regions into four groups, where the conclusions about applicability of the source-strength-constraint approach is different; this also has not been done elsewhere to our knowledge.
- Another approach of this study, which we do not see elsewhere, is the use of a systematic multi-region dataset of satellite snapshots of actual fire cases to evaluate the models, not the monthly means used in most other studies. We believe that comparison of the model instantaneous output matched to the snapshots of actual fires around the globe provides additional unique perspective, which complements the usual model-satellite intercomparison using some averaging spatially and temporally.

We have edited the text to place greater emphasis on these points: rephrased Abstract and Introduction and added a few sentences in section 2.3 under Fig. 2.

- **More of a discussion of MODIS AOD uncertainties would also be useful.**

The MODIS AOD uncertainties have been documented extensively in the literature and some of our coauthors have participated in those studies. However, the key MODIS-measurement-related uncertainty for the current study is not with the AOD, but with the background subtraction. We dealt with this as best we could in Petrenko et al., 2017, and we now elaborate further on the strengths and limitations of what we are able to do given available data in the revised manuscript. We have highlighted that again in the text in the Discussion section.

**Minor:**
- **The intro should make mention of Carter et al. (2020) that looked at fire-influenced AOD in North America.**

We thank you for drawing our attention to this study, and have referenced it in the introduction, also in the discussion of SOA in section 4.2

- **Figure 6 would make more sense if BB was red across both MODIS and model instead of current color scheme.**

We have re-plotted this figure to have all BB AOD as red, and non-BB AOD in blue.

**Review of "Biomass Burning Emissions Analysis Based on MODIS AOD and
AeroCom Multi-Model Simulations", Petrenko et al., 2024, submitted to EGUsphere,**

**I'm rating this paper as "minor revisions", after some waffling on my part between "minor"
and "major". I was concerned that the authors' work did not investigate the variation in
model AOD predictions resulting from the algorithms used in each model to calculate AOD
and the aerosol mixing state properties – there's some previous work where those assumptions
have been shown to contribute up to a factor of 3.5 variation in the resulting AOD estimate
from the model. This is apart from the sources of variability investigated by the authors. My
concern there was that the uncertainty in the conclusion that some of the GFED smoke
emissions values for some subregions were on the low side might be due to underestimates in
the AOD calculation itself (cf. Curci et al., 2015, Atm. Env.). However, once I reached Table 2
in reading through the paper, I could see that the ratio values there suggested underestimates
in model AOD that were outside the range of what might be expected via different aerosol
mixing state assumptions – and I liked the authors approach of caveating their conclusions by
the confidence in the analysis into 4 groups (A,B,C,D).**
**I'm recommending minor revisions with the caveat that the authors add some discussion of the
aerosol mixing state and AOD calculation methodology impact on the results, including that
the analysis of Table 2 and subsequent figures shows that the negative bias AODs calculated by
the models in this work are sufficiently large that AOD calculation methodology can't explain
the results – and hence bolstering the argument that the GFED 3 emissions are likely to be
low. If would also be good to update the references in the introduction (I've given some
examples in my comments below) – though I recognize that a large scale model
intercomparison such as this by default will end up having out of date input information and is
limited by same. For example, GFED 5 came out earlier this year – so its worth a quick
check through the literature and updating the introduction to include some of the more recent
developments.**
**My detailed comments follow, identified by line number. Comments starting with an asterisk
are more serious concerns – which are addressable via more discussion in the text.**

Thank you for your thoughtful and thorough comments!

And thank you for alerting us to the Curci et al., 2015 study. We have now edited the text in
a few places to include mixing state as one of the important parameters for consideration. However,
all the particle microphysical properties relevant to the current study is encapsulated in the model
MEE values, and for the purposes of comparing model with MODIS AOD snapshots, the fact that
the model MEE values vary by a factor 5 is in itself a key result, The details of how the models
arrive at their MEE values is explored in the Curci et al. (2015) paper, but it is beyond the scope of
the current paper. Note also that although we find the GFED BB emissions are underestimated in
some regions (as have previous papers, including Petrenko et al., 2017), the main conclusion of the
current paper is that key model assumptions about MEE, OA/OC, and loss rate, are all too poorly
constrained by observations to consistently apply the source-strength approach to even this well-
studied collection of models. To advance the field, in addition to model sensitivity analyses such as
Curci et al., actual measurements are required, for MEE itself and other factors, but also for the

variables that Curci et al. highlight in their analysis of the ways in which MEE is determined in the models.

Regarding GFED5, at the present only burned area is accessible publicly from the data repository (see https://www.globalfiredata.org/data.html).

**Line 35, 40 Abstract:  There's also the question of how well the model approach for estimating AOD works, which has appeared elsewhere in the literature.  That is, one source of variation is the aerosol mixing state and manner in which BB AOD estimation is carried out, rather than overestimates or underestimates of the amount of smoke.  See Curci et al, 2015, Atm. Env., 115, pp 541-552 (https://www.sciencedirect.com/science/article/pii/S1352231014007018) and note that methods of estimating AOD from model values are generally biased low at 440nm, distributed about the observed values at 870 nm.  One question here is the extent to which the authors results imply that the smoke emissions are biased low versus the methodologies used to estimate AOD in the models are not correct (or not making use of the right information for key parameters such as complex refractive index values)?   Reading through the paper, however, I can see that the overall biases the authors are seeing are larger than the differences that might be associated with AOD calculation methodology – I think some discussion of the Curci paper in a few places in the manuscript would support the argument that the underlying smoke emissions estimates are too low in some regions.**

We do conclude that the difference between the simulated and observed BB AOD is too large to be explained by emissions (or any other single factor) alone. Moreso, we claim that some parameters in the models, such as, for example, MEE or particle mass loading or loss (and thus, lifetime) are currently not possible to constrain due to lack of regular systematic observations of the relevant properties.

**Lines 55 – 64.  This paragraph is missing some of the more recent papers (sometimes by the same authors) and consequently comes across as being a bit out of date.  There's been a lot of work since the papers referenced here.  Some of these may appear later in the paper, but some examples:**
**Anderson et al (GMDD, 2024):  https://gmd.copernicus.org/preprints/gmd-2024-31/; a model specifically designed for real-time forecasting which uses hotspot detection and statistics of burned area per hotspot rather than FRP).**
**Chen et al 2019: GMD 12, 3283-3310, 2019 (same system as above, for North America).**
**Chen et al., 2023:  most recent GFED 5 paper, Earth Science Data Discussions, 1-52, 2023.**
**Van der Werf et al., 2017:  most recent GFED paper (I think - GFED4.1); Earth System Science Data 9(2), 697-720, 2017.**
**Wiedinmyer et al., 2023:  FINN update, EGUsphere, 1-43, 2023.**
**The authors should also be referencing the previous intercomparison of biomass burning by Pan et al 2020:  Atm. Chem. Phys., 20, 969-994, 2020.**
**… all if they haven't already done so.**

All of the above papers now referenced in this paragraph, and several other places in the relevant discussion.

**\*\* Line 79.  The authors should include in their list of things that need to be perfect, "The manner in which models calculate AOD from predicted aerosol fields, and the assumptions made in those calculations regarding aerosol properties such as mixing state and optical properties."**

We have added "and the assumptions made to constrain aerosol properties and processes" to the list of things to consider in line 79.

We have also included mixing state and optical properties among the factors affecting MEE calculation (line 452, 485) and also in Discussion and Conclusions section of the paper.

**There really needs to be some discussion of how the variability in AOD calculations from model output depends on the assumptions regarding the aerosol mixing state, optical properties, etc, at this point.**
**The authors have (largely) missed an important issue here - that there is uncertainty in the AOD calculation itself, quite separate from the models particulate matter estimates.  Issues such as whether the particles' mixing state is a homogeneous versus core-shell can influence the resulting AOD (Curci et al, Atm Env., 115,541-552, 2015).  While models have been underestimating AOD relative to satellite retrievals for some time (e.g. see the Curci et al 2015 paper noted above), the extent to which in the current paper this is the result of underestimates in the models' emissions source term, or in the model outputs and aerosol property assumptions used in the models, is less clear.  The authors need to acknowledge these issues, and discuss the extent to which they may influence the authors' conclusions regarding emissions source strength.  For example, Curci et al (2015) showed that the aerosol phys/chem properties could result in a factor of 2 variation in the calculated AOD at 440nm and 870 nm over the EU, with some calculation methods/mixing state assumptions having much lower biases than others.   The point here is that the extent of this bias will have a strong influence on the inferred emissions strength needed to improve model results.  Later on, though, I note that the range of values in the authors' Table 2 is sufficiently outside of the range of variation that said calculation methodologies can't be the sole contributor to the variability.  Mentioning that range from Curci et al in the context of Table 2 would help bolster the author's suggestion that emitted smoke levels being biased low is at least part of the cause of the low AOD predictions.**

We have inserted Curci et al. (2015) as one of the studies that looked at different factors with model intercomparisons. Here (current line 125):

"AeroCom has a long history of performing multi-model experiments in which certain factors are controlled among the model runs, and comparative analysis yields insights into the impact of different model assumptions and parameterizations  (e.g., Bian et al., 2017; Curci et al., 2015; Gliß et al., 2021; Huneeus et al., 2011; Kim et al., 2019; Kinne et al., 2006; Textor et al., 2006; Tsigaridis et al., 2014; Zhong et al., 2022). These efforts have produced a great many insights into the factors affecting model performance and have made it possible to isolate model-specific factors from issues associated with the external constraints…"

**Line 96: What methodology/optical property assumption was used in Petrenko et al 2012 to generate AOD, and where does that methodology fall with regards to Curci et al.'s comparisons?**

GOCART in all our studies (Petrenko et al, 2012, 2017, and this one) uses externally mixed aerosol. Total AOD is the sum of separately calculated AOD's of individual aerosol species. Thus, it looks from Curci et al.'s analysis that it should be on the higher side (GFEXT configuration in Curci et al.'s table 3 results in highest AOD in their Fig. 4). Model assumptions about mixing state are included in our Table 1.

**Line 127: Minor wording clarification needed here, since the reader has not got that far into the paper: I assume "multi-model" here is referring to the overall modelling framework (global or regional air-quality models) rather than the portion of these models used to estimate AOD?**

AeroCom is a framework in which many global models participate, and multiple models contribute to most experiments organized by AeroCom, which makes both the AeroCom and the experiments within it multi-model. Rephrased these sentences to clarify terminology.

**Line 131: Initialize the models' injection heights only? Or is the mass of emissions also initialized by the satellite data?**

Note that the BBEIH is an independent AeroCom Phase III experiment. Just FYI, in the BBEIH project (i.e., Biomass Burning Emission Injection Height), mass of emissions is prescribed by the GFED4.1s emission inventory, and the vertical distribution of BB emissions is constrained by the MISR plume injection height weighting functions (Val Martin, 2010 https:/doi.org/10/5194/acp-10-1491-2010; Val Martin, 2018 DOI:10.3390/rs10101609). Note that the BB emission mass is not concentrated in one layer but is distributed vertically based on a continuity function. Different from the BBEIH experiment mentioned in the text, the BB emission injection height in our experiment is decided by each model, as given in Table 1.

**Line 134: is the Pan et al manuscript still "in prep" or has it been submitted? 2 years since 2022 now. Should this date be 2024?**

Still in preparation to accommodate the results from additional submissions. We are not including this reference in the final version as it has not been submitted at the time of this review.

**Line 138: GFED3.1 is used as the inventory; for those who are not familiar with GFED, does it also provide injection heights, and if not, how are they calculated by each model? This can be a key issue for AOD: higher injection means faster dispersion and presumably lower downwind AOD values.**

GFED3.1 provides emission amount only; it does not provide injection height. We originally requested that the models distribute emissions in the PBL (defined however each model defines it) but some models still used the default setup in their own models. The details of how they were actually injected in different models is given in Table 1 and the first paragraph of section 2.1.

Although we agree that for some of the larger plumes, downwind dispersion could entail some uncertainty associated with initial plume vertical distribution, we are studying primarily smoke plumes close to the source (see more on how cases were defined and treated in the responses below), where dispersion and particle settling are less likely to be significant.

**\*\*Line 159:, Table S2 is missing important information. Table S2 doesn't mention how the aerosol mixing state has been incorporated into each model's AOD calculation, and needs to do so. For example, are all using Mie scattering; are they core-shell approaches or homogeneous mixtures? Is a black carbon core assumed, etc... Please include another column equivalent to Table 3 of Curci et al 2015 in Table S2, so the reader can see exactly what has been done.**

In this work all the assumptions about particle mixing and optical properties as well as calculation methods are included in the models' MEE term. The overall approach to treating AOD as a product of aerosol mass loading and MEE is described at the beginning of section 4.2.

We mention a few sources of uncertainty for the MEE itself in sections 4.2 and Discussion, now including aerosol mixing state, but taking a more thorough look at those individual factors is beyond the scope of this paper, also because it would require a massive data storage and elaboration effort for all models, which was not possible at this time.

Note that in addition to the lack of adequate observational constraints on MEE itself, there is a similar lack of constraints on the factors required to calculate MEE (e.g., Kahn et al., BAMS, 2017). As such, we really don't have anything to add to the model-based assessment of model-based MEE calculations in papers such as Curci et al. 2015).

We have edited Table 1 in the main text of the paper to contain the factors most relevant to the BB AOD discussion in the text. It now includes aerosol mixing state and size distribution previously listed in Table S2.

**Line 166: The differences in injection height could also be a factor in AOD variation. Have the authors quantified this impact or do other differences control model AOD performance?**
- **What will be the impact of the prescribed injection heights on AOD and hence ideas regarding emissions source strength?**
- **How much variation was present in the different models' estimates of PBL height, and how does that impact AOD calculations?**
- **I'm rather surprised that the models not using the prescribed "inject into the PBL" did not attempt to calculate the height from the model atmospheric thermodynamics (cf. Anderson et al, 2024 reference noted above).**

We agree that injection height might be a factor. However, we are using total-column near-source AOD where the smoke source dominates, at least in the Group A and B cases, so the influence of injection height on AOD is minimized. The impact of injection height on total column AOD is the focus of the AeroCom BBEIH experiment, which is ongoing, and the data are being analyzed (X. Pan et al, manuscript in preparation, 2024). However, in the current study, we did not attempt to quantify the effect of EIH on AOD, and design of our experiment would not allow for a reliable study of this topic.

**Line 181:  I'm surprised (perturbed) that the setup for the study did not require common emissions to be used as a constraint on the models, as has been done in regional air-quality model comparisons such as the Air Quality Model Evaluation International Initiative series of papers, Galmarini et al and other special issue papers in Atm. Env., ACP, etc..  Or do the authors here mean that that an inventory generated from multiple sources was used in common by all participating models?**

Hopefully, we understand the question correctly. As we clearly stated in the manuscript, the BB emissions were prescribed – all participating models were required to use the same BB emissions from GFED3.1. Although emissions from other sources were not prescribed, the model-simulated BB aerosols from the same prescribed BB emissions can be properly intercompared by subtracting non-BB run from BASE run, which is the focus of this paper.

**Line 201-204:  The variation in chosen OA/OC ratios has been shown by the authors to have a large impact on primary OA emissions (a factor of 2.6/1.4 variation) – creating another source of variability.  Can the authors provide some justification regarding why this was not harmonized across models in order to remove this source of uncertainty (or, equivalently, a run could have been done with a common OA/OC ratio used to demonstrate the relative impact)?**

This is an oversight when we proposed this experiment to AeroCom. (Discovering this source of model diversity, at least for us, was part of what we learned in the course of this study.) However, we can normalize the OA variables in Fig. 8 to a fixed OA/OC ratio by dividing the OA global emission, loss rate, load, and BBAOD from the individual OA/OC ratios used in each model to a fixed OA/OC ratio to estimate the model diversity if the same OA/OC ratio were used. Figure R2 below compares the model diversities with modelers' choice of OA/OC ratio (left) and normalized to the same OA/OC ratio (right). In this case, the diversity of emission becomes 1.4% and that for OA BBAOD is reduced by 19.7% to 31.6%. This is quite similar to the case of BC BBAOD in Figure 8a where the BB emission of BC is essentially the same among models.
Added this analysis to section 4.2.

[Figure]

Fig. R2. Model diversity of key parameters determining OA BBAOD with various OA/OC ratios used in each model (left, same as Fig. 8a in the manuscript) and with the same OA/OC ratio (right).

**Line 216: the authors need to provide some more basic information (a few sentences) about the MODIS AOD data, and how it is used. For example, the wavelength of the retrieved AOD, the resolution of the satellite pixels, whether individual AOD pixel values are matched to individual model values at the same time or averages is not clear. With regards to the last of these: I wasn't certain whether the background removal procedure makes use of AOD upwind of the fire locations at the time of the fire to determine background AOD levels for differencing, or whether long-term averages were used. Please clarify; different places in the paper seem to imply different ways the background values were constructed.**

We use MODIS Collection 6 dark target 10-km AOD at 550 nm, where pixels with missing retrievals are filled with scaled MERRAero AOD product (Buchard et al., 2015), which is a reanalysis product with MODIS AOD assimilated into the GEOS/GOCART calculations. Such filling allows to estimate average MODIS AOD over the case box even when thick smoke plumes or partial cloud coverage prevent the native retrieval of satellite product. The details of the filling algorithm are available in Petrenko et al., 2017, which is referenced and also briefly summarized in the current paper.

We have added this clarifying information in section 2.3 (line 212 for wavelength and resolution).

The background AOD value for each case box comes from the MODIS Collection 6 level 3 daily AOD product (MOD08_D3) at 1° horizontal resolution. So, each case box gets one background value. To estimate it, we produce a histogram of all mean 1°-pixel AOD values within the case boundaries from 16 years of Terra observations (2000-2015) for the month at the beginning of or just before the burning season in this region (the month we chose as the background month is listed in Table 4 of P2012). Our reasoning for choosing this month is that, to be ready for the burning, the biomass must have already been growing, so if the biogenic emissions from vegetation

are a significant factor in this ecosystem, they would be accounted for, but before the burning starts, there is no previous smoke contribution. The mode of this histogram defines the most frequent mean AOD within this case box. We call it background, subtract it from all the 10-km pixels within this case box, and what's left we identify as the MODIS BB AOD.

Of course, we are aware of the limitations of this approach due to interannual variability of burning and the variability of the other aerosol sources. To address the need to correct for possible negatives in the BB AOD values obtained this way (not all retrieved 10-km pixels would be above this historically most frequent AOD threshold), we set them to 0 in the BB AOD to have physically meaningful AOD value, but possibly introducing a positive bias into the averaging process. However, even given these limitations of the approach, it was the best way we could come up with to approximate separating BB from non-BB AOD signals. We also tested removing the lowest several % of AOD values within the box, looking upwind or around the delineated smoke plume, but these approaches could not be deployed consistently in all the BB regions because the smoke plume dynamics, shape, burning and transport characteristics are different in different BB regions.

In addition to the Petrenko et al. (2017) reference, we have added more details to section 2.3 to give a better idea of the background-BB aerosol AOD separation process (lines 232-245)

**Line 255:   I'm concerned about how model resolution relative to the size of the box will affect the comparison.  How were the box sizes determined, again?   For example, the highest resolution models will presumably be sampling more grid cells within the prescribed box to generate the average.  These models presumably will also do a better job of resolving the maximum value within the plume.  However, this advantage will be lost if the averaging box is large in size.  I suggest the authors also add the model maximum within the sampling box as well as the average into their comparisons:  perhaps the high resolution models have higher concentrations and hence better AODs locally... but this improvement may not be seen in the averages constructed by the authors.**

Please see the discussion of the sizes and treatment of the case boxes in two questions below.
We considered taking the maximum AOD in each case box, but abandoned this idea because the resolution of the models varies by a factor 2-8 and all models are considerably more coarse than the 10-km MODIS product, so we decided comparing maximum AOD in a box would not be meaningful, and the average over each case domain would be more appropriate.
We also briefly looked at a possible correlation between model resolution and model-satellite average AOD ratio and did not see a pattern.

**Line 268:  empty bullet point in text.  Suggest adding a "box maximum AOD" be added here.**
Removed rogue bullet point.

**Line 273:  The use of averages over a common box size also has inherent risks of smoothing out the model values.  What is the size of the satellite AOD retrieval pixel relative to these boxes, and the size of the model grid cells relative to these boxes?**
We've had many internal deliberations to arrive at the case box as the unit of comparison. Here are some of our considerations, to address your comment:

- The boxes are of different sizes. One of the main criteria for outlining the boxes was that they were at least a 100 km in one linear dimension – so that they are big enough to be resolved by a model with about a 1-degree resolution (main criteria are summarized in section 2.3, lines 215-218) . Most of boxes are larger than this.

- All boxes are near-source. (Fire hot spots must be detected in the MO/YD14 product, and an associated AOD pattern must be observed in the MODIS visible image.) Having smoke cases tied closely in space and time to the actual fire allows for meaningful analysis when the case sizes vary. If the case box encloses the actual smoke plume, then we are directly investigating the model BB processes. For example, for small cases, the emissions in the box will be smaller, and the AOD of the plume will depend on the model dynamics, deposition, chemistry, optical algorithm, as appropriate. For larger cases, the smoke plume associated with the fire will be defined by the same model processes but over a larger burning area, or alternatively, more fires will be included in the box. (There will be more particles in the box from multiple sources, but this plume will also be bigger, so, consistently, AOD will be averaged over the larger box.)

  This near-source approach also calls for model sampling closest to satellite observations. We took AOD from the daily mean of the UTC date of satellite observation (we also did not notice a significant difference between sampling the closest 3-hr model output vs daily output), and other variables (loading, deposition, extinction) from the monthly mean, because that was the model output available.

- Regarding the resolution of the models and satellite product. Participating models have horizontal resolutions from 0.5 x 0.625 to 4 x 5 degrees. With satellite product at 10 km horizontal resolution, we needed some common resolution to compare them. Averaging all these different products over the case box seems reasonable, because it would smooth any possible higher peaks in finer-resolution products, and distributes the variable of interest over the same area.  Also, given different sizes of the case boxes, this approach includes the variable information relevant to every fire case, regardless of its size. So, to get the case average, we average the values of all grid boxes, the centers of which fall within the geographic boundaries of the case rectangle, whatever the original resolution of the grid we are averaging. (We did try to calculate case box averages, taking into consideration partial inclusion of the model grid box or satellite pixel, but it did not make much of a difference, and having the center of the model/satellite grid fall within the boundaries of the fire case is an approximation that includes a large enough portion of the grid to be averaged equally.)

**Line 279, "generally lower BB AOD than the MODIS estimates":  Which might be expected, if AOD is e.g. 440 to 550 nm, based on Curci et al, 2015, and depending on the assumptions used in calculating model AOD.  This does not necessarily imply that the emissions going into the models are low, at this stage in the paper; it could be the result of an inaccurate method for calculating AOD.  Or some of the other assumptions, such as the model emissions being (most models) limited to within the boundary layer - which is not likely to be a limiting factor for large fires.**

See answers to previous questions regarding the AOD calculations w.r.t. mixing state and injection height. We use AOD at 550 nm. Added this information in section 2.3

**\*\*Line 300, Table 2. Up to this point I was thinking that the absence of discussion on aerosol optical depth calculation methodology and assumptions would warrant a "major revisions" for the paper, this Table convinced me it should be minor. There needs to be a bit more discussion of the values shown here relative to variations in Curci et al., 2015. What I can see here is that the ratios show negative biases in excess of the range of values seen in Curci et al, where AOD calculation methodologies and aerosol mixing state properties were compared. Which in turn helps the authors' implication that the analysis is suggesting that some biomass burning emissions levels from GFED are low. The generic model negative biases in some regions (CEAM, TENA, SEAS, BOAS_W, CEAS-E, CEAS-W) are larger than what would be expected associated with choices of model mixing state and the manner in which AOD is calculated. That is, Curci et al showed a generic negative bias factor of 1 to 3.5 in model-generated AOD calculation (their Figure 4) - while the regions in the current authors' work have even further negative biases. The small numbers thus can't be attributed solely to the model assumptions on aerosol mixing state and how they calculate AOD. i.e. the underestimates are outside of the range that might reasonably be associated with AOD calculation methodology, and hence can be attributed to other causes (e.g. emissions magnitude, etc).**
**Similarly, given that Curci et al showed a general underestimate of AOD at shorter wavelengths, the values higher than 1 in this table should be considered "outside of the range that might be expected associated with model AOD calculation variability" and hence may be due to other causes, such as emissions overestimates. Its worth bringing this up in the discussion, since it helps bolster the argument that the analysis is showing an underestimate in smoke emissions magnitude.**

As a follow-up to our response to "Line 96…" comment: Curci et al. (2015) conclude that changing from external to internal mixing decreases AOD. With more than half of the models using the external mixing assumption and generally underestimating satellite-measured AOD in our study, switching to internal mixing could make the discrepancy even larger.

**\*\*Bottom of Table 2: The authors should add another row with the model standard deviation across regions. i.e. some models have an average that appears to be good (close to 1.0), but they also have a large variation between regions. Others may have a consistent relative bias but less variability. Standard deviation would help the reader distinguish the models with "good average performance" from those with "good average performance due to balancing positive and negative biases in some regions".**
We have added a row with the standard deviation of all case ratios per model with the corresponding text in section 3.1

**Suggest the authors add this point to their numbered points in section 3.2. Also, line 348: can a "diversity" row or column be added to Table 2? That would address that concern.**

The bullets in Section 3.2 list the factors considered in classifying regions as being part of Group A, B, C, or D.  Model AOD and BB AOD were sufficient to perform an effective classification; additional aspects of model diversity are considered in the subsequent analysis (e.g., Section 4 and Figure 8)

We have added "StDev" and "Diversity" (calculated as StDev/Mean*100%) columns to Table 2, with corresponding text in 3.1

**Following line 350:  Text needs to include another paragraph with interpretation of Table 3.  What does this Table tell the reader about relative source strength of the emissions (are they biased high or low), for example, and the authors level of confidence in that inference?  How the final column was calculated from the numbers in the Table or other information was not clear; needs a better description in the text.**

This table is a numerical representation of the three factors (numbered list in section 3.2) for multi-factor BB region analysis (we have removed the last factor mentioned in the original submission, because the first three factors are sufficient to separate BB region into groups, and the list is focused on this purpose) . Table 3 is also meant to numerically support the separation of 13 BB regions into four groups according to algorithm in Fig. 5.

The last column was not calculated from the previous numbers in the column, it was the same as the last column in Table 2. Rather each column provides an independent piece of the puzzle, which (columns) need to be looked at in concert to support the algorithm in Fig.5.

This table was one of the versions of the table in which we put together and highlighted different factors to see what patterns emerge. It was originally used to develop the way to segregate the BB regions into groups. We have now changed the formatting and contents of the table, and amended the corresponding text in section 3.2 to make clear how this table supports the story line.

**Table 3:  Table caption needs to include the color coding mentioned in the text.  Not mentioned in the text was the blue values (unless I missed it, apologies if so) - presumably the most uncertain?  Also, there are two different shades of green shown, not described in the text by the time the reader gets to the table.  Was the lighter green intended to represent cases where one of the two sets of values used to get the ratio was "yellow", the other "green", or something else entirely.  The color coding needs to be better described in the Table caption.**

Different greens were a glitch, should have been one color. Now got rid of colors altogether, using bold font instead.

**Table 3:  I'm not sure how the final column was calculated, please explain.  I was trying perturbations of the numbers appearing in the other columns, couldn't end up with the final column ratio values.**

Please see reply above on updates to Table 3.

**Figure 6 discussion could have used a bit more interpretation.  e.g. it looks like BONA, SHAF have a tendency to underestimate AOD from non BB sources while CEAS-E overestimates non**

**BB source AOD.  For BB, BONA, SHAF, NHAF, NHSA, AUST are ok, but rest are biased low.  Relative to Curci et al, BOAS-W, CEAM, TENA, SEAS, CEAS-E, SHSA are probably significant above bias levels that could be associated with the method of AOD calculation, which is an important point to make. Suggest adding something to the figure like a horizontal arrow showing the direction of increasing uncertainty in the interpretation (I got the impression from the previous section that A group is the highest, D group is the lowest; would be nice to have a visual cue on the Figure itself).**

Your impression on the level of uncertainty in groups is correct. However, at this point, the level of our uncertainty in different groups, as described in the text, is qualitative. So, we would not want to over-interpret our results by assigning quantitative significance where we do not have adequate constraints.

**Figure 6b.  Note that from Curci et al, a second region could be drawn, at about 1/3.5 = 0.28 :  values below this line could not be attributed to the model AOD calculation methodology and model assumptions regarding mixing state, assuming AOD is at about 440 nm (somewhere please state the AOD wavelength).  So BOAS_W, CEAM, TENA, SEAS, CEAS_E, CEAS_W.  have biases beyond what might be expected from AOD calculation methodology, which is a useful thing to be able to say.**

There are many factors contributing to model bias. Without additional, designated model experiments, we are not able to quantitatively attribute the model behavior to a particular cause or exclude that cause, such as emission injection height, AOD microphysical and optical properties, aerosol removal processes etc. Therefore, we decided not add a line on Fig 6b as suggested by the reviewer.  At this time, we have added more references and more depth to the discussion of this and other parameters in the Results and Discussion.

**Line 413:  See my above comments regarding the relative role that the AOD calculations and mixing state assumptions might have - this could reduce the list a bit further to include those where the differences are larger than calculation methodology could account for the changes.**

Noted. AOD calculation method and mixing state are among many factors affecting model-satellite AOD comparison, as we conclude further in the paper. Introducing and exploring them is beyond the scope of this project, and is not needed to reach the conclusions presented.

**Line 435:  "properties used in the model."   This doesn't surprise me, given Curci et al; range of a factor of 2 to 3 in model results depending on these assumptions - and added to that, there is the range of emitted values resulting from the OA/OC ratio employed in each model. Can the authors modify the diagram to indicate the relative impact of the OA/OC ratio on the diversity?**

This is addressed in Fig. R2 above, which shows the impact of OA/OC ratio on model diversity. We also appended Fig. 8c (discussed above) to the paper with corresponding text to address this.

**Line 442: "properties used in each model" ... as well as the assumptions used to calculate AOD in each model... should include this. Similarly, line 445, "mixing state" should be added to this list.**

Added mixing state to what was line 445 (now line 452).

**Line 468: Explain how a factor of 2 leads to a diversity of 28% (offline; doesn't have to be in the paper) - I'm curious to see how the calculation works.**

We calculate diversity according to the formula: Diversity = StDev(array)/mean(array)*100%

Here's the original input that led to the numbers in the table. We have updated load calculations for a couple models (just making sure all the averages and outliers are consistently treated across all models and regions, nothing major), so our final numbers in this re-submitted version of the manuscript are slightly different. The Excel formula in the cell O3 is =STDEV(IF(C3:L3<>0,C:L3))/M3*100

| | A | B | C | D | E | F | G | H | I | J | K | L | M | N | O | P |
|---|---|---|---|---|---|---|---|---|---|---|---|---|---|---|---|---|
| 1 | Annual OA BB | budget | GISS-OMA | OsloCTM2 | GISS-MATRIX | GEOSCHEM-v902 | CAM5 | SPRINTARS | GOCART | GEOS | ECMWF-IFS-CY45R: | ECHAM6-SALSA | Mean | Median | Std/mean | max/min |
| 2 | Loss rate | kg/m2/s | 1.9293E-06 | 1.02659E-06 | 1.5969E-06 | 1.72946E-06 | | 1.4487E-06 | 1.6351E-06 | 9.37713E-07 | 0.000002936 | 1.96731E-06 | 1.68966E-06 | 1.63505E-06 | 34.79 | 3.13 |
| 3 | Emission | Tg | 21.97 | 40.82 | 21.97 | 33.08 | | 40.72 | 28.26 | 21.98 | 24.56 | 21.99 | 28.37 | 24.56 | 28.08 | 1.86 |
| 4 | Lifetime | days | 6.00 | 11.27 | 7.25 | 6.69 | | 7.99 | 7.08 | 12.34 | 3.94 | 5.88 | 7.61 | 7.08 | 34.87 | 3.13 |
| 5 | Load | kg/m2 | 8.8216E-07 | 1.2585E-06 | 8.8928E-07 | 1.4204E-06 | 7.3563E-07 | 1.8565E-06 | 1.0776E-06 | 1.34985E-06 | 3.8469E-07 | 7.89749E-07 | 1.06443E-06 | 9.83427E-07 | 39.40 | 4.83 |
| 6 | MEE | m2/kg | 5076.88 | | | 3686.40 | 5279.85 | 5585.72 | 5478.81 | 3526.13 | 4523.70 | 2840.87 | 4499.80 | 4800.29 | 22.91 | 1.97 |
| 7 | OA BBAOD | | 0.00448 | | | 0.00527 | 0.00389 | 0.01022 | 0.00594 | 0.00475 | 0.00173 | 0.00233 | 0.00483 | 0.00461 | 53.88 | 5.89 |
| 8 | | | | | | | | | | | | | | | | |

**Line 470: How many of the models include the formation of secondary organic aerosols, another form of OA that will be created by the organic gases released by fires? Will this influence diversity? See also my comment for line 485, below.**

All emissions saved in all models are for POA. In some cases, the output contains SOA load (CAM5, OsloCTM2, GEOSChem).

We are using total OA variables (making sure that SOA is included in this total where applicable) for BB calculations. For example, GOCART also includes SOA but only biogenic SOA, so when the non-BB run is subtracted from the BB run, other emissions are not included in BB loads and AOD.
We did not request that models save separate output for BB SOA production (in CAM5, OsloCTM2, and GEOS-Chem), such that total source = direct POA emission (prescribed) + SOA production (if it was considered in the model), that would be included as an additional source contributing to the total BB OA emission. So, since we did not save this separate bit of information, it does contribute to the model diversity.

It also came to our attention that the additional SOA source assessment by Carter at al. (2020) concluded that though changing SOA parameterization in the model can lead to a factor of 2 change in BB AOD, both the large possible variation in POA emission estimates and lack of consistent observations to constrain SOA source makes it challenging to estimate the amount of BB SOA.
We added a few sentences about the model differences in total BB OA because of SOA, as well as inclusions/omission of SOA as a potential source of discrepancy in Section 4.2. (lines 508-518)

**Line 474: General comment: It's unfortunate that a scenario with all models being forced to use the same OA/OC ratio was not included! This variation in how the emissions were treated muddies the waters considerably. Why were the models not constrained to use the same values (if only in a scenario) – was this issue not recognized until post-simulations?**

The model diversity from using the same OA/OC ratio can be estimated from the data we have. Please see our response to the "Line 201-204" and Figure R2.

**Line 479: Alternatively, do any of these models include secondary organic aerosol formation? Regional models show this can be a dominant source of OA mass.**
Yes, see above

**Line 485: Ah, there we go… ok, can the relative fraction of SOA/OA in the biomass burning plumes be extracted from these models (models often speciate SOA from primary OA explicitly, so this is something the contributing modellers would have to hand)? The relative fractions would show how much this might influence diversity (and would also be a good point to mention with respect to AOD underestimates – if the SOA/OA fraction is large, and AOD is being underestimated by the models lacking SOA, then adding SOA is a reasonable recommendation.**

We have included the following in the revision (line 513): "In the attempt to work with total OA output provided by the models, whether the model includes SOA simulation or not, these three models also include SOA in their load and loss estimates, with BB SOA contributing around 5% to loads and AOD of BB OA in both CAM5 and OsloCTM2, and 15% in GEOSChem, with these fractions being much smaller than the SOA fraction of total (BB and non-BB) OA; further, these values varying greatly both seasonally and regionally in all the models."

**Line 490: this is about the MEE range you might expect from Curci et al 2015.**
This is good to know that we are coming to the same conclusions by different methods

**Line 495-496 list: Also constrained by the aerosol mixing state assumptions and the extent to which aerosols might be assumed to be core/shell versus homogeneous mixtures ← Add these to the list.**
Added mixing state to the list. All models that use internal mixing assume homogeneous mixing.

**Line 515: Yes – one problem is cases where seasonality of the different sources of aerosols differ. It might be better in that context rather than averaging and generating differences between BB and non-BB runs, to subtract off the background on a finer time scale.**

We did subtract non-BB runs from BASE run in every model time step, no averaging was applied before obtaining model BB variables.

**Line 520: Another possibility that the authors might want to consider (for future work) is using other co-located and co-emitted satellite BB species such as CO emissions to identify the**

**BB cells, and use finer time resolution for removal of background (e.g. background values just before fires used for removal of background during fires).**

Good point. GFED inventories provide a rich selection of such species to use as emissions. But the same uncertainty remains – satellite observed BB CO has to be estimated by subtracting non-BB CO from total CO.

**Line 532: See my earlier comments - a more rigorous constraint based on the variability associated with the methodology for calculating AOD alone could/should be applied here. There are a group of models and regions for which the biases relative to observations are larger than the range of biases Curci et al (2015) saw for AOD at 440 nm. It's the factor of 10 cases (and the ones less 28% or so) that are key.**

Please see our notes on earlier iterations of the same comment above.

**Line 544: Yes - one thing I'm wondering is whether this is an "emission factor" problem (that is, the mass of PM emissions per kg biomass burned is wrong), or if the estimate of the burned area is wrong. Do other emitted species such as CO have the same biases?**

The common method estimating species emission from biomass burning is:

Emission = burned area * fuel load * completeness of burning * species emission factor

Each of the factors on the right side contributes to the uncertainties in estimated emission. Regarding the emission factor, the uncertainty of the emission factor for longer lived gas species, such as CO (lifetime 30-60 days), is significantly lower than that for short-lived species such as aerosol OA and BC (lifetime a few days), because it is more straightforward to evaluate the estimated emissions from measurements of CO, and the atmospheric loss process is much simpler than for aerosols. In our modeling experience, the agreement between model simulated and observed CO is usually much better (e.g., Bian et al., 2010).

Previous studies show that variations/deficits in burned area (BA) is about an order of magnitude larger than that of emission factors (Petrenko et al, 2012, Pan et al, 2020, Van der Werf, 2017), so changes in burned area tend to be the most consequential to changes in emission estimates.

**Line 551-552: Surely some additional constraints such as requiring all models to use the same OA/OC ratio in the emissions should also be mentioned here?**

Yes, it could. We did not do that in this study because it was not in the AeroCom BB experiment protocol, so we accept the model results without unified OA/OC ratio. However, as mentioned earlier, the results of using a same emission factor can be easily obtained by scaling different emission factors from each model to a common value. In the revision, we have provided the expected outcome if a common OA/OC ratio were used.

**Line 570:  Maybe add, "For example, we have shown that the choice of an OA/OC ratio, which varies considerably between the models, can have a significant impact on subsequent AOD predictions."**

    Added

**Line 576:  Mixing state assumptions, etc.  - the manner in which AOD is calculated also contributes to the variability.**

Added "mixing state" to the list of differences between the models

**Reviewer 3:**
**Tables S1 and S2 are useful information. Please consider move it to the main text or Appendix so that it appears in the main paper.**

Table S1 provides general information of model setup for anthropogenic, dust, volcanic, and sea salt emissions that are not the focus of this paper, so we prefer to leave it in the supplement. Most relevant information in Table S2 is now included in the updated Table 1 in the main paper, so we leave Table S2 in the supplement with some modification to avoid repetition.

**Line 88 "Not surprisingly, there are significant discrepancies among the different estimates of BB aerosol source strength" Please add the following reference:**
**Wiedinmyer, C., Kimura, Y., McDonald-Buller, E. C., Emmons, L. K., Buchholz, R. R., Tang, W., Seto, K., Joseph, M. B., Barsanti, K. C., Carlton, A. G., and Yokelson, R.: The Fire Inventory from NCAR version 2.5: an updated global fire emissions model for climate and chemistry applications, Geosci. Model Dev., 16, 3873–3891, https://doi.org/10.5194/gmd-16-3873-2023, 2023.**

Thank you for this reference. We added it in the previous paragraph, along with other references for the bottom-up emission inventories. We tried to limit the list of references after the "not surprisingly…." sentence to the ones intercomparing several emission inventories used by one model (added the newer Carter et al. 2020 reference here.)

**Line 128 and Line 134: This paragraph is a bit confusing. The goal of this paper is to constrain smoke strength? Or inter-model comparison to understand model processes that impact model AOD? Or to evaluate emission inventories and injection height? Or three of them? Did you evaluate injection height in this study? All these processes are subject to uncertainties. In conclusions it says "we explored in some detail the strengths and limitations of the P2017". You also draw a few conclusions for GFED3.1. However you have to assume the P2017 approach is valid in order to draw these conclusions on GFED3.1 and the model performances.**

Originally, in P2017 we thought we could use satellite measurements to constrain model AOD. This worked out reasonably well when we applied the technique to just one model (i.e., P2017). But that was before we attempted the exercise with 11 models, at which point we realized that model-satellite AOD comparison is about so much more than possible emission deficits. As a result, the current study emphasizes *How much more*. We actually do a fairly thorough job of probing the limits of the P2017 approach for multiple models.

We edited the text to clarify the objectives of the study (line 137): "The objectives of this study are: (1) to assess and quantify the AeroCom-model-simulated BB AOD performance as an indication of smoke source-strength provided by the common emissions inventory, (2) to identify regions where the emission inventory might underestimate or overestimate smoke sources based on the comparison between multi-model outputs and the satellite observations, and (3) to assess model diversity and identify underlying causes based on the model-measurement analysis."

We do not evaluate injection height in this study. There is phase 2 of the BB experiment currently under way, led by Xiaohua Pan that is focused on the EIH.

We have expanded the discussion in Section 4.2 on the contributions of several parameters, such as SOA sources, that can contribute to simulated vs observed AOD.

We found it necessary to shift our focus in the discussion and conclusions from GFED3.1to other factors that can produce AOD discrepancies between models and observations and among the models themselves; we conclude that systematic observations are needed to put some constraints on at least some of these parameters.

**I understand it's not possible to re-run all the simulations with GFED4.1s. But please discuss the potential impact of using GFED3.1 rather than GFED4.1s since the two versions are different and GFED4.1s includes more small fires. And you do have low small-fires correction.**

We are aware that GFED4.1s includes small fires lacking in GFED3.1. We addressed the difference in some detail in P2017, and have added the discussion of GFED3.1 vs GFED4s BB in the Discussion section (line 550).

**In the introduction, the description of P2012 and P2017 are too long. Please shorten it and move some content to section 2.**

The current paper builds directly upon the results of P2012 and P2017; the approach, its strengths and limitations, and the underlying MODIS AOD dataset are all developed in those previous papers. As such, it is necessary to provide at least some background on these two papers. Note that comments from other reviewers also required providing more detail on some aspects of our previous papers.

**Figure 5: Are these cases all based on 1-day MODIS AOD dataset? But even for the same region, during different seasons the plume might meet different criteria and be classified to different groups. And even without considering seasonality, are all the plumes in the same region have similar features and fall into same groups?**

The cases are based on the instantaneous snapshots (Level 2 dataset) of actual BB cases, which happen at different times in different regions. The dataset is representative of the seasonality of the biomass burning in the appropriate regions, as reflected in Fig. 2 of the paper. More details on how the cases were defined – the criteria and the process are described in detail in P2017, summarized in the current paper, and for additional considerations please see our response to the comment "Line 273.." by Reviewer #2.

We must rely on statistics to summarize the behavior by region in nearly any global study. That said, we classified the regions based on similar performance and geographic location when applying our source-strength-estimation approach. You will note that we reclassified a couple of regions in the current paper compared to P2017, mainly to better stratify the cases based on their performance. In fact, we first discovered during the study that led to P2012 that within certain geographic regions, fire plumes behaved quite similarly.

**Figure 6: Get rid of the red wavy lines in Figure legend.**

Removed wavy lines

**Line 409: Using GFED3.1 for inter-model comparison study to understand model uncertainties is still reasonable. However, studying the GFED3.1 emission inventory itself seems outdated.**

Fair point. Although this was our original intention, the current paper does not emphasize recommendations for adjusting GFED3; it focuses primarily on the inter-model comparison and highlighting possible sources of model diversity and model-satellite discrepancy.

**Figure 7: add legend.**

Added

**Section 4 is more interesting and I suggest to expand it more and shorten Section 3.**

We have added some results and discussion into section 4 and 5.

**Before doing these analyses, a general evaluation of model AOD from these models are needed. It will be helpful to directly evaluate these models with MODIS AOD product (more than the AOD ratio model/MODIS shown in Figure 6b). This can be included in supplement to at least provide some information on how these models perform.**

General model evaluations with satellite products have been performed by the individual modeling groups as well as a number of model AOD evaluations with MODIS AOD in a number of AEROCOM Phase 2 and Phase 3 studies e.g., Zhong et al. (2022), Gliss et al. (2021), Kinne et al. (2006). We reference these studies in line 125, but believe there is no need to duplicate that work in the current paper.

---

## Referee Report (RR1)

2nd Review of Petrenko et al., 2024, Reviewer 2.

The authors have adequately answered most of my concerns (going through the responses, I've marked most of them with "Ok", "OK, good.", "OK, good point", "Yes, this makes sense", etc). At this point it's down to the "very minor/technical comments".

I should mention that the first paragraph of the response had me wondering whether my suggestions/comments had been followed – though I could see they had been, in the more detailed responses which followed.

The minor technical issues that remain:

(1) Please modify "In addition to the frequent, global AOD and aerosol type that can be provided by satellite aerosol remote-sensing, this necessitates systematic aircraft measurements of detailed microphysical properties for the major aerosol airmass types near-source as well as during transport and aging" in the Conclusions." to replace the words "microphysical properties" to "microphysical and optical properties". I could see this had been done elsewhere in the revised text; the authors might have missed this instance.

(2) The Conclusions sentence "Other aspects of model treatment of aerosol microphysical and optical properties, such as size distributions, mixing states, hygroscopic properties, and MEE will also affect the BB AOD calculations, but these effects are expected to be less significant in the BB source regions" needs either an extension or another line where the authors explain why they believe the last part, "these effects are expected to be less significant in the BB source regions" to be the case. The statement appears to contradict other parts of the modified text, e.g. elsewhere in the Conclusions "This often points to differences in model treatment of physical and chemical processes such as plume injection height, aging time, removal mechanisms, and secondary aerosol formation, as well as aerosol microphysical and optical properties such as particle size distributions, mixing state, hygroscopic growth rates, and mass extinction efficiencies" – which seems to suggest that the latter considerations ->are<- important or could be significant.

(3) Table 1 helps address my concerns about mentioning the optical calculations carried out in the models. Two things worth adding, either in Table 1 or in the accompanying text: (a) was some form of Mie scattering (or a lookup table based on same) used for all models, and (b) please include in the text that the models using internal mixing all assumed a homogeneous mixture (mentioned in the Response to my comments, but not in the revised paper, I think). Also, can the authors please identify which of the references in the right-hand column of Table 1 include the optical calculation details (e.g. identify the optical calculation details reference in italics), for readers who may want to follow up on this.

(4) Page 8 of the Author's response to my comments mentions "...we are studying primarily smoke plumes close to the source (see more on how cases were defined and treated in the responses below), where dispersion and particle settling are less likely to be significant." With regards to particle settling – the authors may want to qualify the portion of the statement about particle settling – this is very dependent on the particle size. e.g. the deposition velocities (last stage of particle settling) for particles > 10 um diameter is 10 to 100 cm s-1. i.e. for the big particles, you have a substantial at-source reduction in

particulate matter (see for example Emerson et al., 2020 PNAS paper). I don't know whether any of the models include large particles, however.

(5) Injection heights – thanks for including the approximations used in Table 1. I'm surprised that none of these calculated plume height explicitly based on the energy release and the temperature lapse rate (cf the Anderson et al 2024 reference). I understand the authors point – at-source, the column total is less likely to be affected by this. I nevertheless wonder about a 6 km high plume versus a 1 km high plume being subjected to very different advection losses at the plume top.

(6) Thanks for the OA/OC discussion in the response and the revised Figure R2. – yes, this sort of thing comes out of doing comparisons like this (agree that's its an important result of the current work, that the OA/OC variation can affect models results).

(7) Line 273 response (page 11): can this portion of the response to the reviewer please be placed in the Supplement and mentioned in a line in the text?

(8) Final suggestion: can the authors please include their definition of Diversity the first time it appears in the revised text. In the original manuscript, it was in the Figure 8 caption (and I missed it, the first time around); their response to my line 468 comment made me do a search on Diversity – and in the revised manuscript it now appears before Figure 8, in Table 2. So defining it the first time it's mentioned, in the text, would help readers to connect the dots, and avoid the confusion I faced in trying to figure out how it was calculated.

---

## Author Response (AR2)

Dear Editors and reviewers,
Our response to the comments are in italics under each respective comment.
Mariya Petrenko and co-authors
* * *
2nd Review of Petrenko et al., 2024, Reviewer 2.
The authors have adequately answered most of my concerns (going through the responses, I've marked most of them with "Ok", "OK, good.", "OK, good point", "Yes, this makes sense", etc.). At this point it's down to the "very minor/technical comments".
I should mention that the first paragraph of the response had me wondering whether my suggestions/comments had been followed – though I could see they had been, in the more detailed responses which followed.
The minor technical issues that remain:

1. (1) Please modify "In addition to the frequent, global AOD and aerosol type that can be provided by satellite aerosol remote-sensing, this necessitates systematic aircraft measurements of detailed microphysical properties for the major aerosol airmass types near-source as well as during transport and aging" in the Conclusions." to replace the words "microphysical properties" to "microphysical and optical properties". I could see this had been done elsewhere in the revised text; the authors might have missed this instance.

*Added "… and optical". Line 610*

2. (2) The Conclusions sentence "Other aspects of model treatment of aerosol microphysical and optical properties, such as size distributions, mixing states, hygroscopic properties, and MEE will also affect the BB AOD calculations, but these effects are expected to be less significant in the BB source regions" needs either an extension or another line where the authors explain why they believe the last part, "these effects are expected to be less significant in the BB source regions" to be the case. The statement appears to contradict other parts of the modified text, e.g. elsewhere in the Conclusions "This often points to differences in model treatment of physical and chemical processes such as plume injection height, aging time, removal mechanisms, and secondary aerosol formation, as well as aerosol microphysical and optical properties such as particle size distributions, mixing state, hygroscopic growth rates, and mass extinction efficiencies" – which seems to suggest that the latter considerations ->are<- important or could be significant.

*Removed "these effects are expected to be less significant in the BB source regions" from the sentence. You are right, there is no evidence in this study to back this up. Line 560*

3. (3) Table1 helps address my concerns about mentioning the optical calculations carried out in the models. Two things worth adding, either in Table 1 or in the

accompanying text: (a) was some form of Mie scattering (or a lookup table based on same) used for all models, and (b) please include in the text that the models using internal mixing all assumed a homogeneous mixture (mentioned in the Response to my comments, but not in the revised paper, I think). Also, can the authors please identify which of the references in the right-hand column of Table 1 include the optical calculation details (e.g. identify the optical calculation details reference in italics), for readers who may want to follow up on this.

*A note on homogeneous mixing and Mie scattering is now included in line 164 of the paper. Regarding the references for calculating optical properties – some models have several references that contain this information, e.g. values table in one, algorithm in the other and an update on it in yet another paper, so there's no uniform answer, and getting this information would require different path for different models. Therefore, we choose to not italicize references relevant for optical properties retrieval at this time.*

4. (4) Page8 of the Author's response to my comments mentions "…we are studying primarily smoke plumes close to the source (see more on how cases were defined and treated in the responses below), where dispersion and particle settling are less likely to be significant." With regards to particle settling – the authors may want to qualify the portion of the statement about particle settling – this is very dependent on the particle size. e.g. the deposition velocities (last stage of particle settling) for particles > 10 um diameter is 10 to 100 cm s-1. i.e. for the big particles, you have a substantial at-source reduction in particulate matter (see for example Emerson et al., 2020 PNAS paper). I don't know whether any of the models include large particles, however.

*BB aerosol particle sizes in all models are smaller than 10 um in diameter. Wildfires can of course produce larger particles, but they tend to represent a very small fraction of the total optical depth. So, we did not include a discussion of larger particles in the current study.*

5. (5) Injection heights – thanks for including the approximations used in Table 1. I'm surprised that none of these calculated plume height explicitly based on the energy release and the temperature lapse rate (cf the Anderson et al 2024 reference). I understand the authors point – at-source, the column total is less likely to be affected by this. I nevertheless wonder about a 6 km high plume versus a 1 km high plume being subjected to very different advection losses at the plume top.

*We are not including any comment on advection losses or other effects of injection height in the paper. The calculation of plume height from first principles is very approximate, due to large uncertainty in the dynamical heat flux at the source (often taken as 5 or 10 times the observed 4-micron brightness temperature anomaly – i.e., the FRP or fire radiative power to produce generally realistic heights), and in entrainment caused by turbulent mixing during plume-rise is almost entirely unconstrained (e.g., Kahn et al., JGR 2007, and many others). Further, although injecting emissions at 1 km and at 6 km should have*

*different advection and loss processes, our focus is the BB emission at the source locations such that the investigation of injection-height observations, and the resulting conclusions that could be drawn are beyond the scope of this study.*

6. (6) Thanks for the OA/OC discussion in the response and the revised Figure R2.–yes, this sort of thing comes out of doing comparisons like this (agree that's its an important result of the current work, that the OA/OC variation can affect models results).

*Thank you.*

7. (7) Line 273 response (page 11): can this portion of the response to the reviewer please be placed in the Supplement and mentioned in a line in the text?

*Yes. Added a Supplement 1 to the Supplemental material "On the use of case box as a unit of spatial and temporal comparison". Made a note about it in the text – line 275*

8. (8) Final suggestion: can the authors please include their definition of Diversity the first time it appears in the revised text. In the original manuscript, it was in the Figure 8 caption (and I missed it, the first time around); their response to my line 468 comment made me do a search on Diversity – and in the revised manuscript it now appears before Figure 8, in Table 2. So defining it the first time it's mentioned, in the text, would help readers to connect the dots, and avoid the confusion I faced in trying to figure out how it was calculated.

*Added definition of diversity to line 307*